# Glia-neuron coupling via a bipartite sialylation pathway promotes neural transmission and stress tolerance in *Drosophila*

**Hilary Scott[1†‡], Boris Novikov[1†], Berrak Ugur[2§], Brooke Allen[1], Ilya Mertsalov[1#], Pedro Monagas-Valentin[1], Melissa Koff[1], Sarah Baas Robinson[3], Kazuhiro Aoki[3], Raisa Veizaj[4], Dirk J Lefeber[4], Michael Tiemeyer[3], Hugo Bellen[2], Vladislav Panin[1]\***

[1]Department of Biochemistry and Biophysics, Texas A&M University, College Station, United States; [2]Departments of Molecular and Human Genetics and Neuroscience, Baylor College of Medicine, and Jan and Dan Duncan Neurological Research Institute, Texas Children's Hospital, Houston, United States; [3]Complex Carbohydrate Research Center, University of Georgia, Athens, United States; [4]Translational Metabolic Laboratory, Department of Neurology, Donders Institute for Brain, Cognition and Behavior, Radboud University Medical Center, Nijmegen, Netherlands

**\*For correspondence:**
panin@tamu.edu

[†]These authors contributed equally to this work

**Present address:** [‡]Ocular Genomics Institute, Massachusetts Eye and Ear, Boston, United States; [§]Departments of Neuroscience and Cell Biology, Yale University School of Medicine, New Haven, United States; [#]Institute of Developmental Biology, Russian Academy of Sciences, Moscow, Russia

**Competing interest:** The authors declare that no competing interests exist.

**Abstract** Modification by sialylated glycans can affect protein functions, underlying mechanisms that control animal development and physiology. Sialylation relies on a dedicated pathway involving evolutionarily conserved enzymes, including CMP-sialic acid synthetase (CSAS) and sialyltransferase (SiaT) that mediate the activation of sialic acid and its transfer onto glycan termini, respectively. In *Drosophila*, *CSAS* and *DSiaT* genes function in the nervous system, affecting neural transmission and excitability. We found that these genes function in different cells: the function of *CSAS* is restricted to glia, while *DSiaT* functions in neurons. This partition of the sialylation pathway allows for regulation of neural functions via a glia-mediated control of neural sialylation. The sialylation genes were shown to be required for tolerance to heat and oxidative stress and for maintenance of the normal level of voltage-gated sodium channels. Our results uncovered a unique bipartite sialylation pathway that mediates glia-neuron coupling and regulates neural excitability and stress tolerance.

## Editor's evaluation

This important paper uses *Drosophila* as a model to study the sialylation pathway and its role in nervous system function. Intriguingly, the authors demonstrate that the final two steps of the sialylation biosynthetic pathway are split across glia (CSAS) and neurons (DSiaT). This compelling finding will interest a broad readership as it identifies a new mode by which glia support neuronal function.

## Introduction

Protein glycosylation, the most common type of posttranslational modification, plays numerous important biological roles, and regulates molecular and cell interactions in animal development, physiology, and disease (*Varki, 2017*). The addition of sialic acid (Sia), i.e., sialylation, has prominent effects due to its negative charge, bulky size, and terminal location of Sia on glycan chains. Essential roles of sialylated glycans in cell adhesion, cell signaling, and proliferation have been documented in many studies (*Schwarzkopf et al., 2002*; *Varki, 2007*; *Varki, 2008*). Sia is intimately involved in the function

of the nervous system. Mutations in genes that affect sialylation are associated with neurological symptoms in human, including intellectual disability, epilepsy, and ataxia due to defects in sialic acid synthase (N-acetylneuraminic acid synthase [NANS]), sialyltransferases (ST3GAL3 and ST3GAL5), the CMP-Sia transporter (SLC35A1), and CMP-Sia synthase (CMAS) (*Hu et al., 2011*; *Mohamed et al., 2013*; *Boccuto et al., 2014*; *van Karnebeek et al., 2016*). Polysialylation (PSA) of NCAM, the neural cell adhesion molecule, one of the best studied cases of sialylation in the nervous system, is involved in the regulation of cell interactions during brain development (*Schnaar et al., 2014*). Non-PSA-type sialylated glycans are ubiquitously present in the vertebrate nervous system, but their functions are not well defined. Increasing evidence implicates these glycans in essential regulation of neuronal signaling. Indeed, N-glycosylation can affect voltage-gated channels in different ways, ranging from modulation of channel gating to protein trafficking, cell surface expression, and recycling/degradation (*Cronin et al., 2005*; *Watanabe et al., 2007*; *Ednie and Bennett, 2012*; *Baycin-Hizal et al., 2014*; *Watanabe et al., 2015*; *Thayer et al., 2016*). Similar effects were shown for several other glycoproteins implicated in synaptic transmission and cell excitability, including neurotransmitter receptors (reviewed in *Scott and Panin, 2014*). Glycoprotein sialylation defects were also implicated in neurological diseases, such as Angelman syndrome and epilepsy (*Isaev et al., 2007*; *Condon et al., 2013*). However, the in vivo functions of sialylation and the mechanisms that regulate this posttranslational modification in the nervous system remain poorly understood.

*Drosophila* has recently emerged as a model to study neural sialylation in vivo, providing advantages of the decreased complexity of the nervous system and the sialylation pathway, while also showing conservation of the main biosynthetic steps of glycosylation (*Koles et al., 2009*; *Scott and Panin, 2014*). The final step in sialylation is mediated by sialyltransferases, enzymes that use CMP-Sia as a sugar donor to attach Sia to glycoconjugates (*Figure 1*; *Varki et al., 2015a*). Unlike mammals that have 20 different sialyltransferases, *Drosophila* possesses a single sialyltransferase, DSiaT, that has significant homology to mammalian ST6Gal enzymes (*Koles et al., 2004*). The two penultimate steps in the biosynthetic pathway of sialylation are mediated by sialic acid synthase (also known as NANS) and CMP-sialic acid synthetase (CSAS, also known as CMAS), the enzymes that synthesize sialic acid and carry out its activation, respectively (*Varki et al., 2015a*). These enzymes have been characterized in *Drosophila* and found to be closely related to their mammalian counterparts (*Kim et al., 2002*; *Viswanathan et al., 2006*; *Mertsalov et al., 2016*). In vivo analyses of DSiaT and CSAS demonstrated that *Drosophila* sialylation is a tightly regulated process limited to the nervous system and required for normal neural transmission. Mutations in *DSiaT* and *CSAS* phenocopy each other, resulting in similar defects in neuronal excitability, causing locomotor and heat-induced paralysis phenotypes, while showing strong interactions with voltage-gated channels (*Repnikova et al., 2010*; *Islam et al., 2013*). *DSiaT* was found to be expressed exclusively in neurons during development and in the adult brain (*Repnikova et al., 2010*). Intriguingly, although the expression of *CSAS* has not been characterized in detail, it was noted that its expression appears to be different from that of *DSiaT* in the embryonic ventral ganglion (*Koles et al., 2009*), suggesting a possibly unusual relationship between the functions of these genes. Here, we tested the hypothesis that *CSAS* functions in glial cells, and that the separation of *DSiaT* and *CSAS* functions between neurons and glia underlies a novel mechanism of glia-neuron coupling that regulates neuronal function via a bipartite protein sialylation.

Glial cells have been recognized as key players in neural regulation (reviewed in *Volterra and Meldolesi, 2005*; *Bosworth and Allen, 2017*; *Magistretti and Allaman, 2018*). Astrocytes participate in synapse formation and synaptic pruning during development, mediate the recycling of neurotransmitters, affect neurons via $Ca^{2+}$ signaling, and support a number of other essential evolutionarily conserved functions (reviewed in *Neniskyte and Gross, 2017*; *Bittern et al., 2021*; *Nagai et al., 2021*). Studies of *Drosophila* glia have revealed novel glial functions in vivo (reviewed in *Freeman, 2015*; *Rittschof and Schirmeier, 2018*; *Bittern et al., 2021*). *Drosophila* astrocytes were found to modulate dopaminergic function through neuromodulatory signaling and activity-regulated $Ca^{2+}$ increase (*Ma et al., 2016*). Glial cells were also shown to protect neurons and neuroblasts from oxidative stress and promote the proliferation of neuroblasts in the developing *Drosophila* brain (*Bailey et al., 2015*; *Liu et al., 2015*; *Kanai et al., 2018*). The metabolic coupling between astrocytes and neurons, which is thought to support and modulate neuronal functions in mammals (*Magistretti and Allaman, 2018*), is apparently conserved in flies. Indeed, *Drosophila* glial cells can secrete lactate and alanine to fuel neuronal oxidative phosphorylation (*Volkenhoff et al., 2015*; *Liu et al., 2017*).

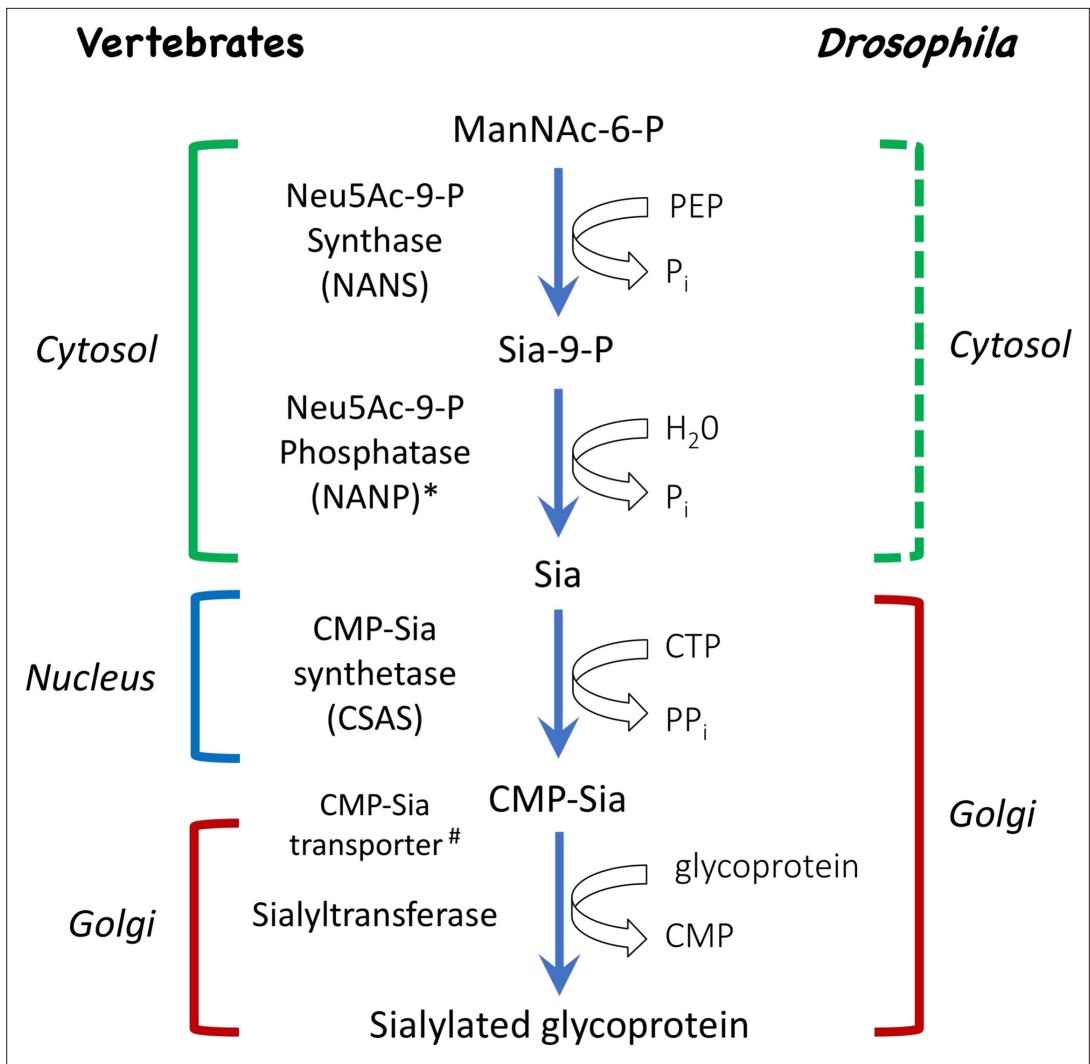

**Figure 1.** Schematic of the sialylation pathways in vertebrate and *Drosophila*. In vertebrates, phosphorylated sialic acid is produced by *N*-acetylneuraminic acid synthase (Neu5Ac-9-P synthase, or NANS) from *N*-acetyl-mannosamine 6-phosphate (ManNAc-6-P), converted to sialic acid by *N*-acylneuraminate-9-phosphatase (Neu5Ac-9-P phosphatase, or NANP), and then activated by CMP-sialic acid synthetase (CSAS, also known as CMAS) to become CMP-Sia, the substrate for sialyltransferase enzymes that work in the Golgi and attach sialic acid to termini of glycan chains. While NANS and NANP work in the cytosol, CSAS enzymes normally localize to the nucleus in vertebrate cells, and the transfer of CMP-Sia to the Golgi requires CMP-Sia transporter. In *Drosophila*, both CSAS and sialyltransferase are localized in the Golgi, and CMP-Sia transporter is not required for sialylation. Biochemical activities of NANS, CSAS, and the *Drosophila* sialyltransferase (DSiaT) were confirmed in vitro (***Kim et al., 2002***; ***Koles et al., 2004***; ***Viswanathan et al., 2006***; ***Mertsalov et al., 2016***). Dashed bracket: subcellular localization was not experimentally confirmed. #CMP-Sia transporter was not identified in invertebrates. *NANP was found to be not essential for sialylation (***Willems et al., 2019***).

In the current work, we described a novel mechanism of glia-neuron coupling mediated by a unique compartmentalization of different steps in the sialylation pathway between glial cells and neurons in the fly nervous system. We explore the regulation of this mechanism and demonstrate its requirement for neural functions.

## Results

### Expression of *Drosophila CSAS* is restricted to glial cells and does not overlap with *DSiaT* expression

Previous studies indicated that *CSAS* is expressed in the nervous system and functions together with *DSiaT* in a pathway that affects neural transmission (*Islam et al., 2013*). However, the expression of CSAS has not been characterized in detail. To determine the expression of CSAS in different cells, we created a LexA reporter construct based on a genomic *BAC* clone that included the *CSAS* gene along with a large surrounding genomic region (*Venken et al., 2009*, see Materials and methods). We modified the *BAC-CSAS* by replacing part of the *CSAS* coding region with the sequence encoding LexA::p65 transcription activator (*Pfeiffer et al., 2010*) using recombineering (*Venken et al., 2009*) to generate a *CSAS-LexA* driver. This strategy has been useful to generate reporters with expression patterns that correspond to endogenous genes (*Venken et al., 2009*). We combined *CSAS-LexA* with *LexAop2-mCD8-GFP* and *LexAop-GFP.nls* reporters to label cell surfaces and nuclei of *CSAS*-expressing cells, respectively, and analyzed the expression pattern of *CSAS* at different developmental stages. Double-labeling experiments using Repo as a glial marker revealed that *CSAS* is expressed in many glial cells in the CNS throughout development and in adult flies (*Figure 2A–B*); in contrast, no expression was detected in neurons (*Figure 2C*). This is a surprising result, considering that DSiaT, the enzyme that functions downstream of CSAS in the sialylation pathway, is expressed only in neurons but not in glial cells (*Repnikova et al., 2010*). To confirm that *CSAS-LexA* expression recapitulates the endogenous expression of *CSAS*, we carried two sets of control experiments. First, we introduced the original *BAC-CSAS* clone as a transgene in flies and combined it with *CSAS* knockout, which resulted in full rescue of the temperature-sensitive (TS) paralysis phenotype of *CSAS* mutants (*Figure 2—figure supplement 1*). This supported the notion that the genomic clone includes all important regulatory elements to induce *CSAS* in endogenous manner. Second, we used *CSAS-LexA* to drive the transgenic expression of *CSAS* cDNA in *CSAS* mutants. This also fully rescued the TS paralysis, supporting the notion that *CSAS-LexA* recapitulates the *CSAS* endogenous expression (*Figure 2—figure supplement 2*). To further confirm that the expression of *CSAS* is indeed confined to the cells without DSiaT expression, we carried out double-labeling experiments to visualize CSAS and DSiaT-expressing cells simultaneously. To label DSiaT-expressing cells, we used a transgenic *BAC-DSiaT-HA* construct carrying a large genomic locus including the *DSiaT* gene modified with a 3xHA tag sequence to allow immunodetection (see Materials and methods). In agreement with previous studies (*Repnikova et al., 2010*), the expression of DSiaT-HA was detected only in differentiated neurons labeled by Elav, but not in neural progenitors expressing a neuroblast marker Deadpan (*Figure 2—figure supplement 3*). Importantly, we observed no overlap between the expression patterns of *CSAS* and *DSiaT* (*Figure 2D–F*). Taken together, these results show that CSAS and DSiaT are expressed in distinct cell populations within the CNS, glial cells, and neurons, respectively.

### CSAS is required in glial cells, but not in neurons

To investigate the cell-specific requirement of CSAS in the nervous system, we carried out rescue experiments using UAS-GAL4 ectopic expression system (*Brand et al., 1994*). CSAS function was shown to be required for normal neural transmission, while *CSAS* mutations cause locomotor defects and TS paralysis phenotype (*Islam et al., 2013*). Using cell-specific GAL4 drivers, we induced the transgenic expression of *UAS-CSAS* in *CSAS* homozygous mutants and assayed them for TS paralysis. Glial-specific expression of transgenic *CSAS* using drivers expressed in all glial cells (*Repo-Gal4*), ensheathing (*Mz709-Gal4*), astrocyte-like (*dEAAT1-Gal4*), neuropile ensheathing glia (*R56F03-Gal4*), or subperineurial glia (*Gli-Gal4*) could fully rescue the phenotype of *CSAS* mutants (*Auld et al., 1995*; *Ito et al., 1995*; *Sepp et al., 2001*; *Rival et al., 2004*; *Doherty et al., 2009*; *Kremer et al., 2017*), while the expression in neurons using a pan-neuronal driver (*C155-GAL4*) or other neuronal drivers broadly expressed in the nervous system (*Mj85b-Gal4, 1407*-Gal4) (*Lin and Goodman, 1994*; *Dubnau et al., 2001*; *Kraft et al., 2016*) did not result in rescue (*Figure 3A*, *Figure 3—figure supplements 1–2*). Interestingly, perineurial driver (*R85G01-Gal4*, *Kremer et al., 2017*) could partially rescue the phenotype, even though perineurial glia is separated from the brain by a tightly sealed layer of subperineurial cells maintaining the blood-brain barrier, and thus perineurial cells are not well poised to provide CMP-Sia for brain functions. This partial rescue is potentially explained by the fact that *R85G01-Gal4* was found to be also expressed in a small number of cortex and astrocyte-like glial cells

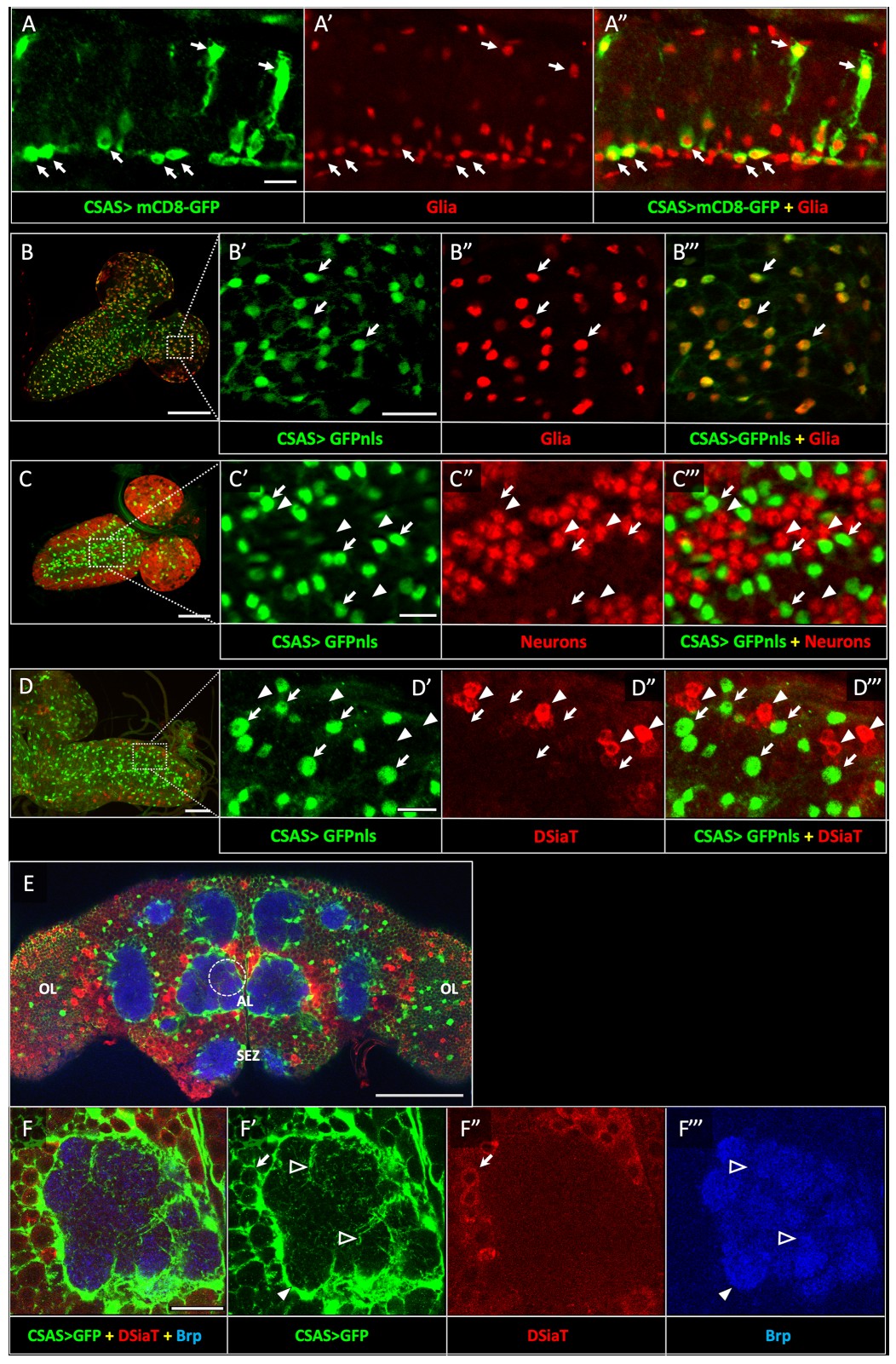

**Figure 2.** *CSAS* expression is restricted to glial cells and shows no overlap with the expression of *DSiaT* during development and in the adult brain. (**A–A″**) *CSAS* expression (green) is detected in glial cells (Repo, red) of the developing ventral ganglion during late embryonic stages. Arrows indicate examples of glial cells with *CSAS* expression. A″ is the overlay of green (**A**) and red (**A′**) channels. (**B–B‴**) *CSAS* expression (green) is present in

*Figure 2 continued on next page*

*Figure 2 continued*

the majority of glial cells (Repo, red) in the CNS at larval stages. (**B′–B‴**) are zoomed-in images of a brain region outlined in B, B‴ is overlay of B′ and B″. (**C–C‴**) CSAS expression (green) is not detected in neurons (Elav, red) in the CNS at larval stages. Arrows and arrowheads indicate examples of cells with *CSAS* and Elav expression, respectively. (**C′–C‴**) are zoomed-in images of a ventral ganglion region outlined in C, C‴ is overlay of C′ and C″. (**D–D‴**) *CSAS* expression (green) is not detected in the CNS cells expressing DSiaT (red) at larval stages. Arrows and arrowheads indicate examples of cells with *CSAS* and DSiaT expression, respectively. (**D′–D‴**) are zoomed-in images of a ventral ganglion region outlined in D, D‴ is overlay of D′ and D″. (**E**) *CSAS* (green) is expressed throughout the adult brain, including glial cells in the optic lobes (OL), around the antennal lobe (AL), and the sub esophageal zone (SEZ), but *CSAS* expression is not detected in DSiaT-expressing neurons (red). (**F–F‴**) Zoomed-in images of the antennal lobe region indicated by a dashed circle in E. *CSAS*-expressing cells produce processes surrounding the soma of DSiaT-expressing projection neurons (arrow), enveloping the antennal lobe (filled arrowhead), and sending fine projections inside the glomeruli (empty arrowheads). Brp staining (blue) labels neuropil in E–F. (A) Embryonic stage 17, lateral view, anterior is left, ventral is up; (B) third instar larval stage, anterior is top-right; (C) first instar larval stage, anterior is top-right; (D) third instar larval stage, anterior is left; (E–F) adult brain, frontal view. Scale bars: 10 µm (**A, C′, D′**), 100 µm (**B, E**), 20 µm (**B′, F**), 50 µm (**C–D**). *CSAS* expression was visualized using *CSAS-LexA* driver-induced expression of GFP with membrane (mCD8-GFP) or nuclear localization (GFPnls) tags. Images were acquired using confocal microscopy.

The online version of this article includes the following source data and figure supplement(s) for figure 2:

**Figure supplement 1.** Rescue of the temperature-sensitive (TS) paralysis phenotype of *CSAS* mutants by *BAC-CSAS*.

**Figure supplement 2.** Rescue of the temperature-sensitive (TS) paralysis phenotype of *CSAS* mutants by *CSAS-LexA*-induced transgenic expression of *CSAS* cDNA.

**Figure supplement 3.** DSiaT-HA expression is detected in neurons but not in neuroblasts.

**Figure supplement 1—source data 1.** Source data for *Figure 2—figure supplement 1*.

**Figure supplement 2—source data 1.** Source data for *Figure 2—figure supplement 2*.

---

(*Weiss et al., 2022*). Taken together, our results demonstrated that *CSAS* expression in glial cells, but not in neurons, is sufficient to restore neural function in *CSAS* mutants.

Sialylation mutants have locomotion defects, such as an inability to promptly right themselves after falling upside down (*Repnikova et al., 2010*; *Islam et al., 2013*). Expression of *UAS-CSAS* in glial cells using *Repo-Gal4* rescued this locomotion phenotype of *CSAS* mutants, while expression in neurons using *C155-Gal4* did not result in rescue (*Figure 3B*). To assess the requirement of CSAS in synaptic transmission, we examined the function of motor neurons using electrophysiological assays at the neuromuscular junctions (NMJs). In sialylation mutants, larval motoneurons exhibit defects in excitability associated with a pronounced decrease of excitatory junction potentials (EJP) at NMJs (*Repnikova et al., 2010*; *Islam et al., 2013*). We analyzed EJPs in *CSAS* mutant third instar larvae and assessed the cell-specific requirement of *CSAS* by transgenic rescue. We used *Repo-Gal4* and *C164-Gal4* (*Choi et al., 2004*) drivers to induce the expression of *UAS-CSAS* in glial cells and motoneurons of *CSAS* mutants, respectively. *CSAS* expression in glial cells was sufficient to restore normal EJPs, however, *CSAS* expressed in motoneurons did not rescue neurotransmission defects (*Figure 3C*). These results provide compelling evidence that CSAS normally functions in glial cells but not in neurons, consistent with the results of behavioral assays.

We also examined the cell-specific requirement of *CSAS* by downregulating its function in different cells. To this end, we knocked down *CSAS* specifically in glial cells or neurons by expressing *UAS-CSAS-RNAi* using *Repo-Gal4* or *C155-Gal4*, respectively. To potentiate the *RNAi*-mediated knockdown, we co-expressed *UAS-CSAS-RNAi* with *UAS-Dcr-2* (*Dietzl et al., 2007*) and used a genetic background that was heterozygous for a *CSAS* deletion allele (*Islam et al., 2013*). The knockdown of *CSAS* in glial cells resulted in the TS paralysis phenotype similar to that of *CSAS* null mutants. No paralysis was induced by knocking down *CSAS* in neurons (*Figure 3D*). Taken together, our data show that CSAS is necessary and sufficient in glial cells to support normal neural functions.

## DSiaT is required in neurons

Previous studies using an endogenously expressed tagged version of DSiaT demonstrated that DSiaT could be detected in neurons but not in glial cells (*Repnikova et al., 2010*). However, whether DSiaT

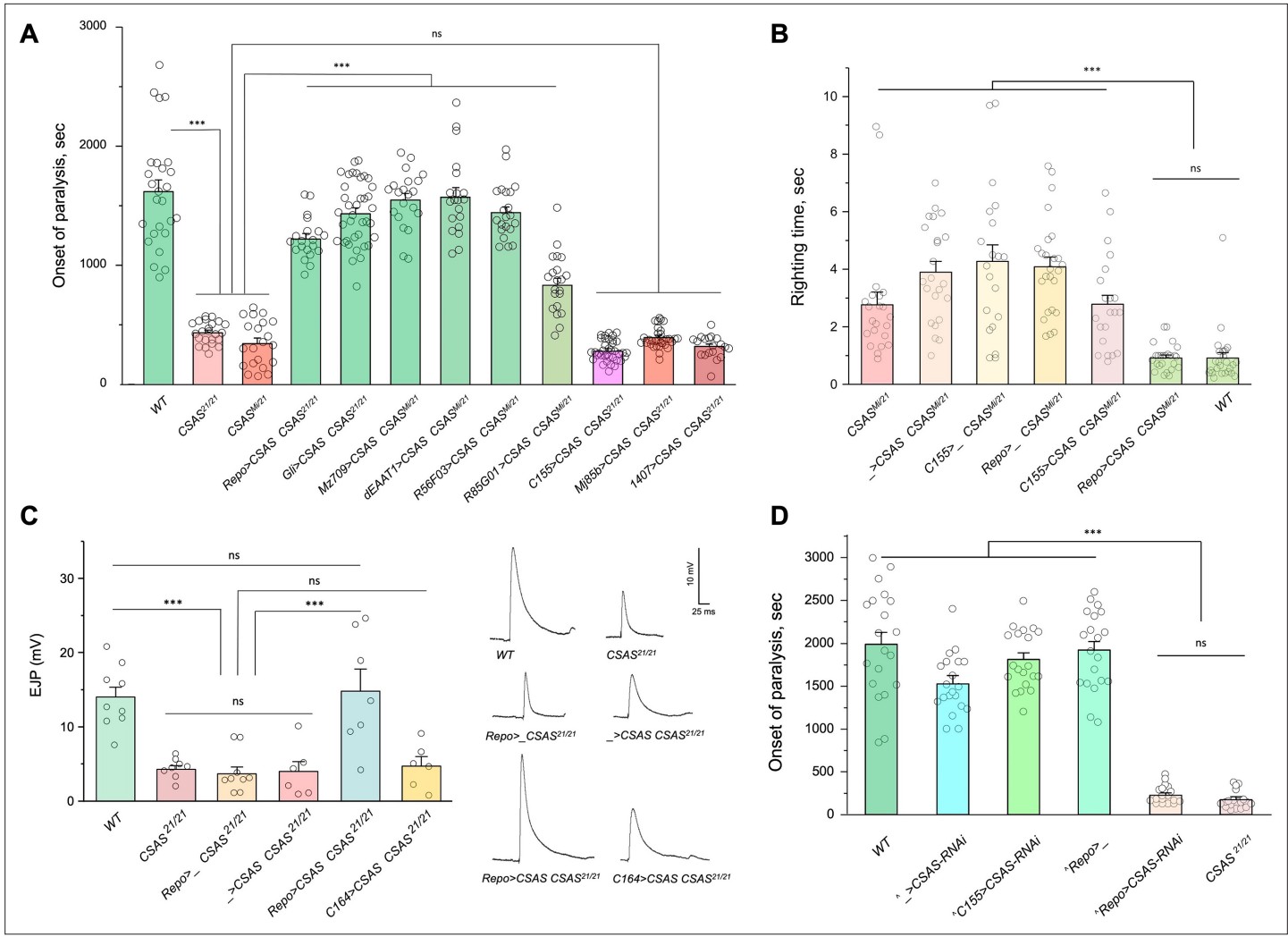

**Figure 3.** CMP-sialic acid synthetase (CSAS) is required in glial cells, but not in neurons, for normal neural functions. (**A**) Rescue of TS paralysis phenotype of *CSAS* mutants using *UAS-Gal4* system. *CSAS*[21] (null) and *CSAS*[Mi] (strong loss-of-function) mutant alleles were used in homozygous and heteroallelic combinations. The expression of transgenic *UAS-CSAS* construct was induced using a panel of cell-specific *Gal4* drivers. A pan-neuronal *Gal4* driver (*C155*) or drivers broadly expressed in the CNS neurons (*1407* and *Mj85b*) did not induce rescue, while the expression of *UAS-CSAS* by glial-specific drivers, including *Repo*, *Gli*, and *Mz709* (expressed in nearly all glial cells, ensheathing glial cells, and subperineural glia, respectively), rescued the phenotype. Analyses of control mutant genotypes (*UAS-CSAS* without driver, and driver-only mutant genotypes) confirmed the specificity of the rescue results (*Figure 3—figure supplement 1*). The expression of CSAS induced by *C155* was confirmed using immunostaining (*Figure 3—figure supplement 2*). At least 19 files (5-day-old females) were assayed for each genotype. (**B**) Locomotor phenotype of *CSAS* mutants rescued using *UAS-Gal4* system. The expression of *UAS-CSAS* induced in neurons by *C155* driver did not rescue the phenotype, while the glial-specific expression driven by *Repo-Gal4* resulted in full rescue. Mutant genotypes with *UAS-CSAS* alone or drivers alone were used as controls, and they did not show rescue. At least 20 females were assayed for each genotype. (**C**) Rescue of neuromuscular excitatory junction potential (EJP) defect of *CSAS* mutants using *UAS-Gal4* system. The reduced EJP phenotype was rescued by glial-specific expression of *UAS-CSAS* induced by *Repo-Gal4*. The expression of *UAS-CSAS* in motoneurons using *C164-Gal4* did not result in rescue. Representative EJP traces are shown on the right. EJPs were evoked in 0.5 mM $Ca^2$ and analyzed at muscle 6/7 neuromuscular junctions (NMJs) of third instar larvae (see Materials and methods for details). 6-9 larvae were assyed for each genotype. (**D**) Cell-specific RNAi-mediated knockdown reveals that *CSAS* is required in glial cells. *UAS-CSAS-RNAi* was induced in glial cells by *Repo-Gal4,* which resulted in TS paralyses phenotype. The expression of *UAS-CSAS-RNAi* in neurons induced by *C155-Gal4* did not cause the phenotype. To potentiate the effect of *CSAS-RNAi*, the knockdown experiments were performed in the genetic background with co-expression of *UAS-dcr2* and heterozygous for *CSAS*[21] mutant allele (^, genotypes with matching genetic background including *UAS-dcr2* and *CSAS*[21/+]). At least 20 females were assayed for each genotype (all data points represent different flies). In all panels: error bars are SEM; one-way ANOVA with post hoc Tukey test was used for statistical analyses; *** $p < 0.001$; ns, no significant difference ($p > 0.05$). See *Supplementary file 1* for detailed genotype information.

The online version of this article includes the following source data and figure supplement(s) for figure 3:

**Source data 1.** Source data for *Figure 3A*.

*Figure 3 continued on next page*

*Figure 3 continued*

**Source data 2.** Source data for *Figure 3B*.

**Source data 3.** Source data for *Figure 3C*.

**Source data 4.** Source data for *Figure 3D*.

**Figure supplement 1.** Temperature-sensitive (TS) paralysis assays of control genotypes for cell type-specific rescue of *CSAS* mutants (*Figure 3A*).

**Figure supplement 1—source data 1.** Source data for *Figure 3—figure supplement 1*.

**Figure supplement 2.** Transgenic expression of *UAS-CSAS* induced by *C155-Gal4* driver in larval brains assayed by immunostaining.

is required specifically in neurons was not examined. Although this question can be in principle clarified by a rescue strategy using *UAS-Gal4* system, this approach has been hampered by the 'leaking' expression of *UAS-DSiaT* that was able to rescue the *DSiaT* mutant phenotypes without the presence of a *Gal4* driver. As an alternative approach, we investigated the cell-specific requirement of DSiaT by *RNAi*-mediated knockdown. To increase the efficiency of knockdown, we carried out the knockdown in heterozygotes for a *DSiaT* null allele, *DSiaT$^{S23/+}$*, which did not show the TS paralysis phenotype themselves. When *DSiaT* was downregulated by the expression of *UAS-DSiaT-RNAi* in neurons using *C155-GAL4*, the flies became paralytic at elevated temperature, showing the TS paralysis, a phenotype that recapitulated that of *DSiaT* null mutants. In contrast, *DSiaT* knockdown in glial cells did not cause the mutant phenotype (*Figure 4*). These results show that DSiaT function is required in neurons, consistent with the expression pattern of DSiaT.

## *CSAS* is required for the biosynthesis of CMP-Sia in *Drosophila*, while both *CSAS* and *DSiaT* are necessary for the production of sialylated N-glycans in vivo

Genetic and phenotypic analyses previously demonstrated that *CSAS* and *DSiaT* genes work in the same functional pathway affecting neural transmission (*Repnikova et al., 2010*; *Islam et al., 2013*). Although the biochemical activities of their protein products were characterized in vitro (*Koles et al., 2004*; *Mertsalov et al., 2016*), the roles of these genes in sialylation were not examined in vivo. Considering the unusual separation of *CSAS* and *DSiaT* expression patterns at the cellular level, we decided to test their requirements for the biosynthesis of sialylated glycans in vivo. First, we analyzed the production of CMP-Sia in wild-type flies and *CSAS* mutants by a liquid chromatography-mass spectrometry approaches (see Materials and methods). A prominent peak corresponding to CMP-Sia was detected in wild-type flies as shown before (*van Scherpenzeel et al., 2021*), while no CMP-Sia was found in *CSAS* mutants. Transgenic rescue using *UAS-GAL4* system resulted in the restoration of CMP-Sia biosynthesis in the mutants (*Figure 5A–B*). These results revealed that the production of CMP-Sia in *Drosophila* specifically requires CSAS activity. Second, we examined N-glycans in *CSAS* and *DSiaT* mutants by mass spectrometry. Sialylated glycans are present in *Drosophila* at extremely low levels (*Aoki et al., 2007*; *Koles et al., 2007*). We decided to focus our analyses on third instar larval brains because *CSAS* and *DSiaT* show prominent expression during late larval stages (*Figure 1B–D* and *Repnikova et al., 2010*; *Islam et al., 2013*). We found N-glycan profiles were dominated by high- and pauci-mannose glycans in all genotypes, with hybrid and complex structures representing a small portion of the total N-glycome (*Figure 5C–D*), consistent with previous studies that analyzed N-glycans produced in embryos and adult heads (*Aoki et al., 2007*; *Koles et al., 2007*). Sialylated structures were detected in wild-type larval brains, but were not detected in *CSAS* or *DSiaT* mutants (*Figure 5C–D*). These results demonstrated that *CSAS* and *DSiaT* are essential for the biosynthesis of sialylated N-glycans in vivo, and that each of these genes plays a non-redundant role in this pathway.

## Biosynthesis of Sia is downregulated in neurons

The ectopic expression of CSAS induced by *C155-Gal4* is predicted to generate CMP-Sia, the sugar donor required for DSiaT, in neurons. However, even though DSiaT is endogenously expressed and functions in neurons, the neuronal expression of *UAS-CSAS* could not rescue the phenotype of *CSAS* mutants (*Figure 3A–B*). This unexpected result may indicate that the sialylation pathway is blocked in neurons upstream of the CSAS-mediated step. For instance, neurons may have a limited capacity to synthesize Sia, a CSAS substrate, due to low activity of NANS (*Figure 1*), the evolutionarily conserved

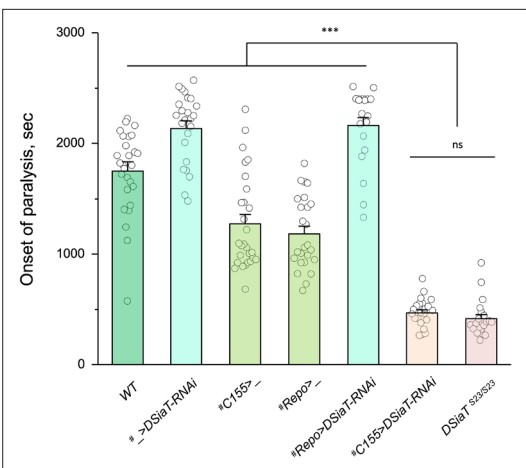

**Figure 4.** Cell-specific knockdown revealed that *DSiaT* is required in neurons. *UAS-DSiaT-RNAi* was induced in neurons by *C155-Gal4,* which resulted in temperature-sensitive (TS) paralyses phenotype. The expression of *UAS-DSiaT-RNAi* in glial cells by *Repo-Gal4* did not produce the phenotype. To potentiate the effect of *RNAi*, knockdown was carried out in the genetic background heterozygous for *DSiaT* mutant allele (*DSiaT^{S23/+}*) and flies were reared at 29°C. 20-28 five-day-old female flies were assayed for each genotype. #, genotypes with matching genetic background heterozygous for *DSiaT^{S23}*. Error bars are SEM; one-way ANOVA with post hoc Tukey test was used for statistical analyses; *** p<0.001; ns, no significant difference (p>0.05).See **Supplementary file 1** for detailed genotypes.

The online version of this article includes the following source data for figure 4:

**Source data 1.** Source data for **Figure 4**.

Sia synthase previously shown to be required for sialylation in *Drosophila* cultured cells (**Granell et al., 2011**). To test this hypothesis, we co-expressed *UAS-NANS* and *UAS-CSAS* transgenic constructs in neurons of *CSAS* mutants. Unlike the neuronal expression of *UAS-CSAS* alone, this co-expression could significantly rescue the phenotype of *CSAS* mutants (**Figure 6**). Taken together, these results indicated that the biosynthesis of Sia is indeed blocked in neurons due to a low level of endogenous NANS activity.

## Paralysis phenotype is highly sensitive to the level of CSAS

The facts that CSAS expression is restricted to glial cells while the biosynthesis of Sia is downregulated in neurons indicate that sialylation is tightly controlled in the nervous system, and that CSAS can potentially play a key regulatory role in the sialylation pathway. To shed light on this possibility, we decided to test how sensitive the neural functions are to heat stress conditions at different levels of *CSAS* activity. To this end, we assayed TS paralysis phenotype of flies with varied levels of CSAS, including *CSAS* null mutants, heterozygous mutants, wild-type, and CSAS overexpression genotypes. We found that the paralysis phenotype is highly sensitive to the level of CSAS activity. *CSAS* homozygous mutants exhibited the strongest phenotype, while the paralysis phenotype of *CSAS* heterozygotes was intermediate between that of homozygous mutants and wild-type flies with matching genetic backgrounds (**Figure 7A**). Remarkably, flies with *CSAS* transgenic overexpression were more tolerant to heat shock than wild-type counterparts (**Figure 7B**). This additional protection from heat-induced stress by upregulation of CSAS suggests that the endogenous level of CSAS may be a limiting factor in the sialylation. Together, these data support the hypothesis that sialylation can play a regulatory role in modulating neuronal transmission and promoting the stability of neural signaling during stress conditions.

## CSAS activity protects from oxidative stress

Oxidative stress is known to affect the nervous system in various ways, with neuronal excitability abnormalities being among the immediate consequences of ROS overproduction (**Wang et al., 2011**). Considering that sialylation is involved in the regulation of neuronal excitability, we decided to test if the sialylation pathway plays a role in ameliorating the effect of oxidative stress. Using a paraquat-induced oxidative stress paradigm (see Materials and methods), we tested the viability of flies with different levels of CSAS activity, including *CSAS* mutant, rescue, wild-type, and overexpression genotypes. *CSAS* mutants were highly sensitive to oxidative stress as compared to matching 'wild-type' control. The glial-specific rescue resulted in significantly decreased mortality of mutants. Moreover, the overexpression of CSAS in glial cells of wild-type flies provided additional protection from oxidative stress, further increasing survivorship (**Figure 8A–B**). These results demonstrated that CSAS is required for protection from oxidative stress and suggested that the modulation of CSAS can underlie an endogenous mechanism that helps maintain neural functions during stress conditions.

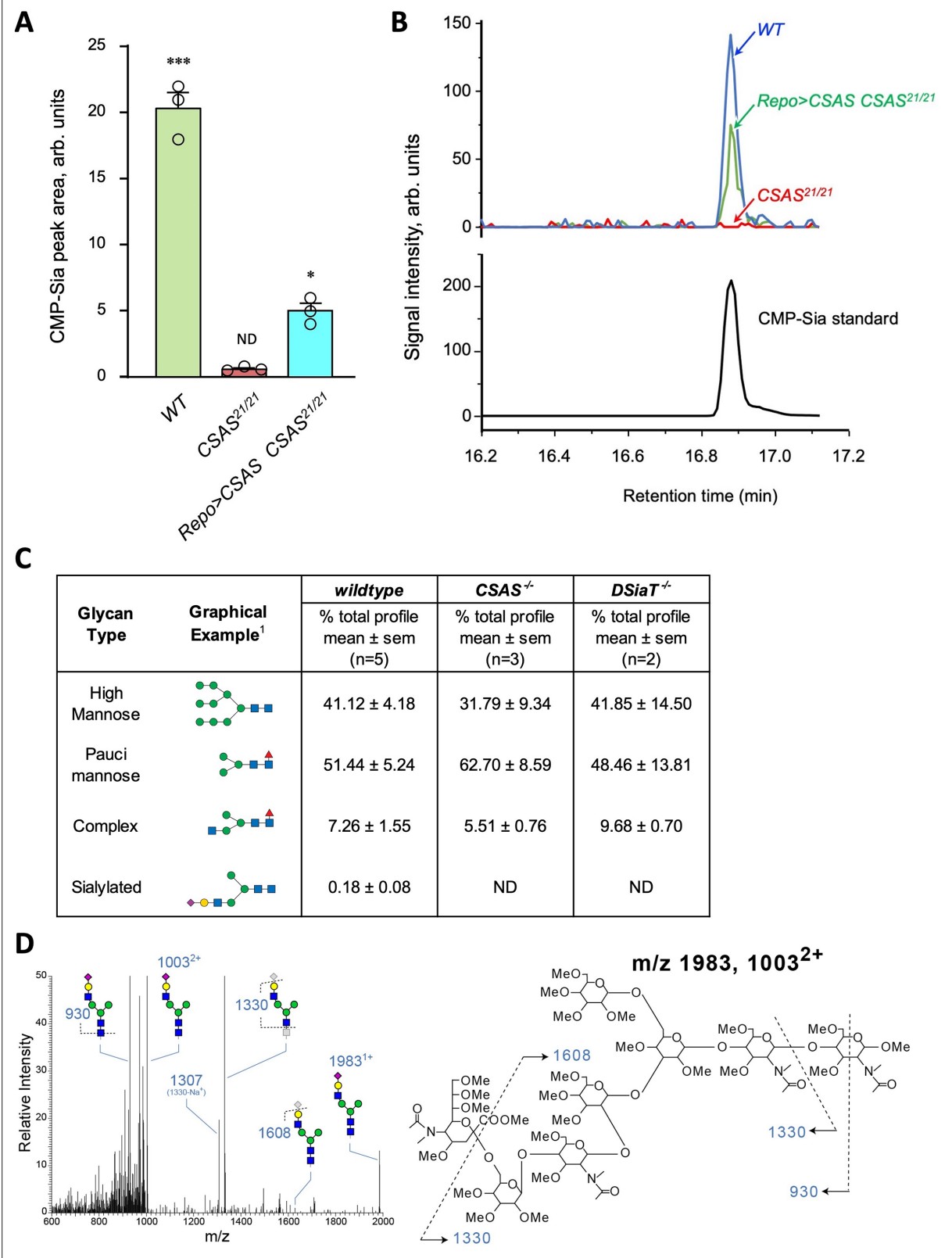

**Figure 5.** Analysis of CMP-Sia and sialylated glycans in *Drosophila*. (**A**) Quantification of CMP-Sia using LC-MS/MS by normalized peak area (see Materials and methods). CMP-Sia was detected in wild-type flies (*WT*) but not in *CSAS* mutants (*CSAS²¹/²¹*). Transgenic expression of *UAS-CSAS* in glial cells of *CSAS* mutants by *Repo-Gal4* (*Repo>CSAS CSAS²¹/²¹*, a rescue genotype) could significantly restore the level of CMP-Sia. ND, not detected (signal/noise ratio <1). Data were obtained from three biological replicates per genotype, each including 100 adult flies (50 males plus 50 females)

*Figure 5 continued on next page*

*Figure 5 continued*

analyzed in three technical repeats. Error bars are SEM; one-way ANOVA with post hoc Tukey test was used for statistical analyses; ***, *, differences with $p < 0.001$ and $p < 0.05$, respectively. (**B**) Typical examples of normalized CMP-Sia signal intensity traces for wild-type, *CSAS* mutant, and rescue genotypes, as well as CMP-Sia standard. (**C**) Summary of glycomic analyses of N-linked glycans in wild-type *Drosophila*, *CSAS*, and *DSiaT* mutants. The N-glycome of third instar larval brains was analyzed. No sialylated N-glycans were detected in the mutants. Samples from wild-type and mutant genotypes were analyzed in parallel using the glycomic protocol described in Materials and methods. n, number of replicates. [1]Most abundant glycan detected in wild-type is shown as representative. ND, not detected. Graphical representation and description of structures are according to the accepted glycan nomenclature (***Aoki et al., 2007***; ***Varki et al., 2015b***). See extended table of N-glycan species identified by glycomic analyses in ***Figure 5—source data 3***. (**D**) Example of fragmentation of a sialylated N-glycan extracted from wild-type larvae. MS/MS fragmentation of the doubly charged, permethylated ion at m/z=1003 (m+Na)$^{2+}$ reveals signature ions consistent with loss of charge (m/z=1983), loss of sialic acid (m/z=608, 1330, 1307), as well as cross-ring fragmentation and loss of reducing terminal residues. The fragmentation pattern confirms the presence of the depicted sialylated structure. Similar fragmentation was not detected in *DSiaT* or *CSAS* mutants.

The online version of this article includes the following source data for figure 5:

**Source data 1.** Source data for ***Figure 5A***.

**Source data 2.** Source data for ***Figure 5B***.

**Source data 3.** Extended table of N-glycan species identified by glycomic analyses in *wild-type*, *DSiaT* mutant, and *CSAS* mutant larval brains.

## Sialylation is required to maintain Para expression

Previous studies found that *DSiaT* and *CSAS* are required for normal neuronal excitability and revealed strong synergistic interactions of these genes with *para*, the *Drosophila* gene encoding voltage-gated sodium channel (***Repnikova et al., 2010***; ***Islam et al., 2013***). These results suggested that the level of Para is potentially affected by sialylation. To test this hypothesis, we employed *para-GFP*, a *Para* allele that endogenously expresses a functional GFP-tagged version of the channel (***Ravenscroft et al., 2020***). The expression of Para-GFP was analyzed in *DSiaT* mutants using western blots. We analyzed flies on day 7 after eclosion, at the age when the TS paralysis phenotype of sialylation mutants becomes prominent (***Repnikova et al., 2010***; ***Islam et al., 2013***). The level of Para-GFP was decreased in *DSiaT* mutants, as compared to a matching 'wild-type' control genotype, while the transgenic expression of *DSiaT* could restore the level of Para-GFP in the mutants (***Figure 9***). To test if this effect on Para could be due to changes in gene expression, we examined *para* mRNA by qRT-PCR. The level of *para* mRNA was not affected in *DSiaT* mutants, which indicated that the effect is posttranscriptional (***Figure 9—figure supplement 1***). As a control for conceivable non-specific effects on membrane proteins in *DSiaT* mutants, we also analyzed the expression of an irrelevant GFP-tagged membrane protein expressed in neurons (mCD8-GFP). The expression of the control protein was not affected in *DSiaT* mutants (***Figure 9—figure supplement 1***), which supported the hypothesis that DSiaT is specifically required to maintain the normal level of voltage-gated sodium channels in neurons.

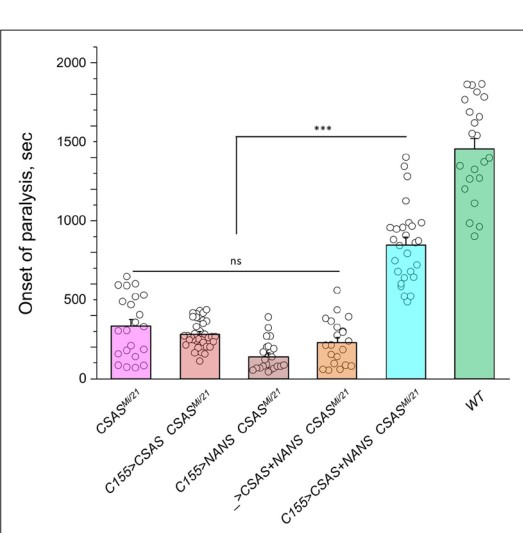

**Figure 6.** Transgenic co-expression of *N*-acetylneuraminic acid synthase (NANS) and CMP-sialic acid synthetase (CSAS) in neurons rescued the phenotype of *CSAS* mutants. *UAS-CSAS* and *UAS-NANS* were expressed individually or together in neurons of *CSAS*$^{Mi/21}$ mutants using *C155-Gal4*. The co-expression of *CSAS* and *NANS* could rescue the temperature-sensitive (TS) paralysis phenotype, while their individual expression did not result in rescue. 22-37 five-day-old female flies were assayed for each genotype. Error bars are SEM; one-way ANOVA with post hoc Tukey test was used for statistical analyses; *** $p < 0.001$; ns, no significant difference ($p > 0.05$). See ***Supplementary file 1*** for detailed genotypes.

The online version of this article includes the following source data for figure 6:

**Source data 1.** Source data for ***Figure 6***.

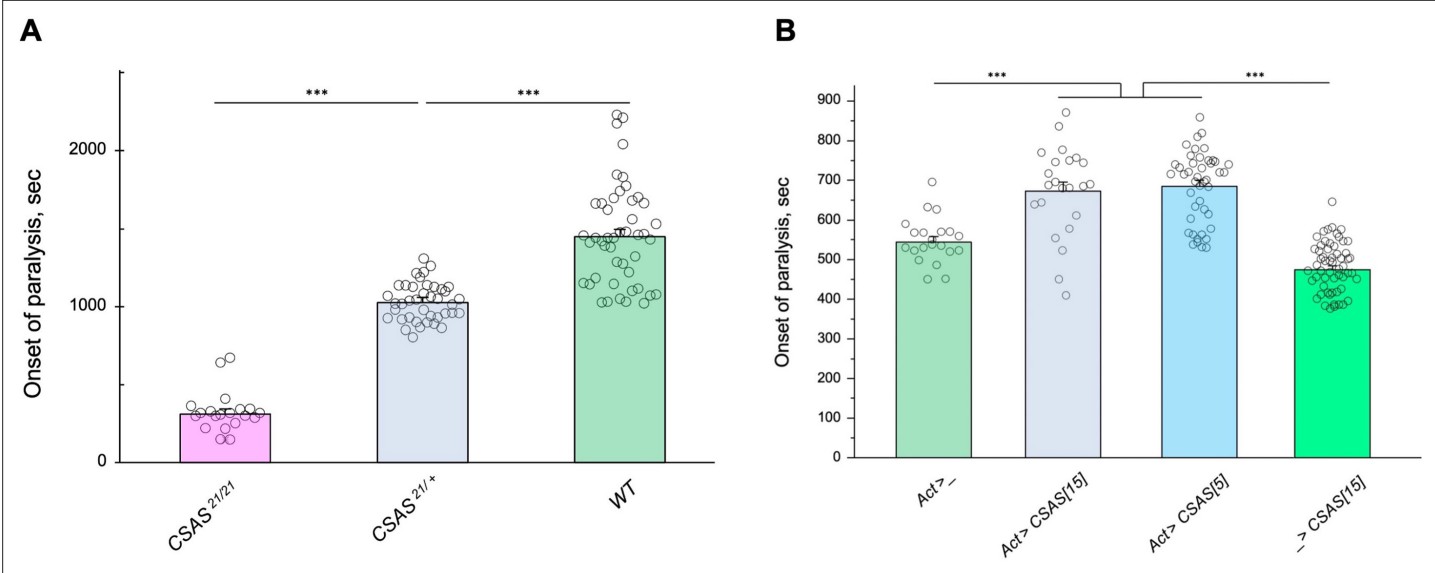

**Figure 7.** Tolerance to heat shock is very sensitive to the level of CMP-sialic acid synthetase (CSAS) activity. (**A**) Comparing phenotypes of *CSAS* homozygous null mutants (*CSAS[21/21]*), *CSAS* mutant heterozygotes (*CSAS[21/+]*), and wild-type flies. Temperature-sensitive (TS) paralyses assays were performed at 38°C. All genotypes had matching genetic backgrounds: the *CSAS[21/21]* mutants were outcrossed to wild-type flies (*WT*) at least seven times; the heterozygotes were obtained from the cross between *WT* and *CSAS[21/21]*. (**B**) Overexpression of CSAS increases tolerance to heat shock. Driver-only genotype (*Act>_*) was used as a 'wild-type' control with a matching genetic background. Two independent *UAS-CSAS* transgenic insertions on different chromosomes (designated as *CSAS[5]* and *CSAS[15]*) were tested to confirm the specificity of the effect. No-driver control (*_>CSAS[15]*) confirmed that the effect is indeed due to CSAS overexpression. TS paralyses assays were performed at 40°C to decrease the time to paralysis. All genotypes were multiply outcrossed to the same *WT* genetic background (*w[1118] Canton S*). (A–B) At least 25 five-day-old females were assayed for each genotype. Error bars are SEM; one-way ANOVA with post hoc Tukey test was used for statistical analyses; *** p<0.001. See **Supplementary file 1** for detailed genotypes.

The online version of this article includes the following source data for figure 7:

**Source data 1.** Source data for **Figure 7A**.

**Source data 2.** Source data for **Figure 7B**.

## Discussion

Glia cooperate with neurons via several evolutionarily conserved functional pathways to provide vital support and regulation of neural functions (*Nagai et al., 2021*). A major mechanism that controls the activity of neural circuits is mediated by astrocyte-mediated uptake and recycling of excitatory and inhibitory neurotransmitters, which is conserved from flies to mammals (*Ma et al., 2016*). Glia can also provide a metabolic support of neurons, which is another important example of glia-mediated effect on neuronal functions. Here, we described glia-neuron coupling that controls neural transmission and relies on the separation of enzymatic steps of the sialylation pathway between neurons and glia (*Figure 10*). We found that DSiaT and CSAS, the enzymes mediating the last two essential biosynthetic steps in the sialylation pathway, are expressed in the *Drosophila* nervous system in separate, non-overlapping cell populations, neurons and glial cells, respectively. DSiaT was previously shown to be expressed in numerous CNS neurons during development and at the adult stage, including larval motoneurons and interneurons, and optic lobe neurons and projection neurons in the adult brain, while no DSiaT expression was detected in glial cells (*Repnikova et al., 2010*). Here, we revealed that *CSAS* expression is present in many cells throughout the CNS starting from late embryonic stages, but this expression was confined to a subpopulation of glial cells (*Figure 2A–C*). Different types of glial cells showed *CSAS* expression, including astrocytes, cortex, and neuropile ensheathing glia (*Figure 2F*). No expression was detected in neurons at any developmental stage. This was confirmed by double-labeling experiments with the pan-neuronal marker Elav that demonstrated no overlap between CSAS and Elav expression patterns. These data are consistent with recently published single-cell transcriptomic data of the third instar and adult brains (*Ravenscroft et al., 2020*; *Li et al., 2022*). Accordingly, no overlap in expression pattern was found between *CSAS* and *DSiaT* known to be

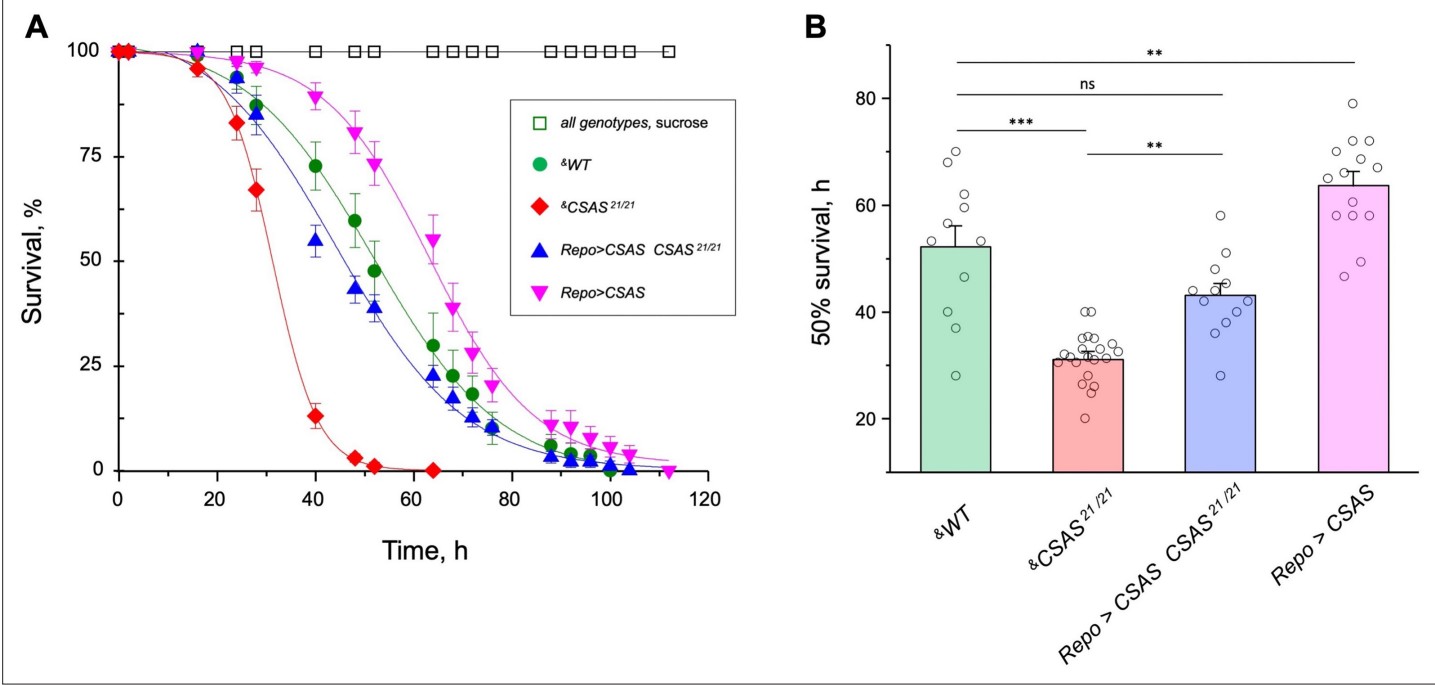

**Figure 8.** CMP-sialic acid synthetase (CSAS) affects tolerance to oxidative stress. (**A**) Survival in oxidative stress conditions was assayed using a paraquat exposure paradigm. *CSAS* homozygous mutants (*CSAS²¹/²¹*), *CSAS* mutant rescue (*Repo>CSAS CSAS²¹/²¹*), wild-type (*WT*), and transgenic overexpression (*Repo>CSAS*) genotypes were exposed to 40 mM paraquat or sucrose as a control. Every genotype was assayed on paraquat using >10 independent biological replicates, each including a group of 8–12 flies; at least 50 flies were assayed for sucrose control. &, all genotypes had matching genetic backgrounds that included *Repo-Gal4* driver. All mutant alleles and transgenic insertions were outcrossed to the same WT background multiple times. Statistical significance was analyzed by (i) log-rank test of cumulative survival data for each genotype (*Figure 8—figure supplement 1*) and (ii) one-way ANOVA with post hoc Tukey test applied to survival at 40, 48, and 52 hr. These analyses revealed significant differences between all genotypes (p<0.05), except for *WT* and rescue that were not always significantly different from each other. Error bars are SEM. (**B**) Comparison of 50% survival time on paraquat estimated from the data shown in A. Error bars are SEM. One-way ANOVA with post hoc Tukey test was used for statistical analyses; *** p<0.001; ** p<0.01; ns, no significant difference (p>0.05). See ***Supplementary file 1*** for detailed genotypes.

The online version of this article includes the following source data and figure supplement(s) for figure 8:

**Figure supplement 1.** The Kaplan-Meier survival curves from paraquat exposure experiments.

**Figure supplement 1—source data 1.** Source data for *Figure 8—figure supplement 1*.

**Source data 1.** Source data for *Figure 8A*.

**Source data 2.** Source data for *Figure 8*.

expressed exclusively in neurons (*Figure 2D*). Such a strict separation of expression at the cellular level suggested that there are distinct cell-specific requirements for *DSiaT* and *CSAS* functions. We confirmed this conclusion by genetic manipulations of these genes' activities using mutant alleles and transgenic expression constructs. These experiments demonstrated that *CSAS* is necessary and sufficient in glial cells, while *DSiaT* is expressed and required in neurons (*Figures 3–4*). Despite such an unusual partitioning of *DSiaT* and *CSAS* functions at the cellular level, our biochemical analyses revealed that these genes play essential non-redundant roles in the biosynthesis of sialylated N-glycans in vivo (*Figure 5*). Interestingly, the transgenic expression of CSAS in neurons could not rescue *CSAS* mutants despite the fact that CMP-Sia, a product of CSAS, is expected in that case to be efficiently utilized in neurons by endogenous DSiaT. This result suggested that the biosynthetic pathway of sialylation might be blocked in neurons upstream of the CSAS-mediated step. Indeed, this conclusion was supported by transgenic co-expression of CSAS and NANS in neurons of *CSAS* mutants, which resulted in mutant rescue (*Figure 6*). This indicated that the pathway is specifically blocked in neurons due to the insufficient activity of *NANS*, the gene encoding an evolutionarily conserved enzyme that generates Sia using phosphoenolpyruvate (PEP) and *N*-acetyl-mannosamine 6-phosphate as substrates (*Figure 1*). Although little is known about the function of NANS in vivo, the enzymatic activity of *Drosophila* NANS was confirmed in vitro (***Kim et al., 2002***; ***Granell et al., 2011***). Previous

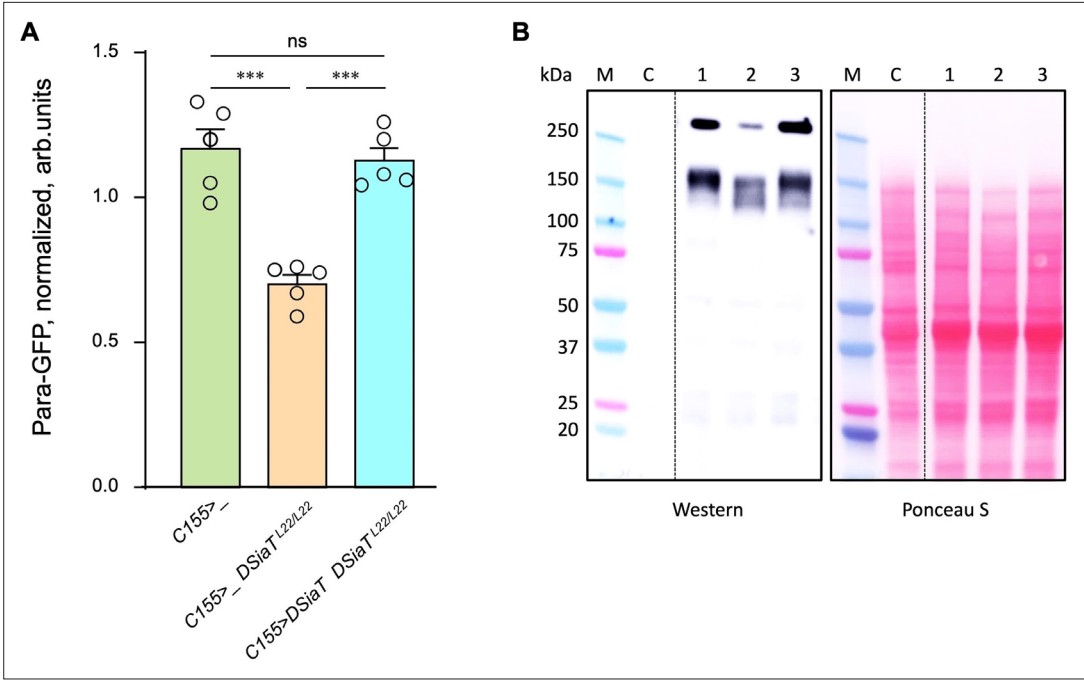

**Figure 9.** The level of Para is decreased in *DSiaT* mutants. (**A**) Quantification of Para-GFP by western blots revealed that *DSiaT* mutants (*C155>_ DSiaT^{L22/L22}*) have a lower level of Para-GFP as compared to a control 'wild-type' genotype (*C155>_*). The level of Para-GFP is restored in the mutants by transgenic expression of *UAS-DSiaT* using *C155-Gal4* driver (*C155>DSiaT DSiaT^{L22/L22}*, rescue genotype), which confirmed the specificity of the phenotype. Para-GFP signal was normalized to total protein amount analyzed by Ponceau S staining. The analysis is based on five biological replicates (data points shown), each including up to three technical repeats (see Materials and methods for details). Error bars are SEM. One-way ANOVA with post hoc Tukey test was used for statistical analyses; *** p<0.001; ns, no statistical difference (p>0.05). (**B**) A representative example of Para-GFP western blot and corresponding Ponceau S protein staining. M, molecular weight marker; C, control flies without Para-GFP; 1, *C155>_*; 2, *C155>_ DSiaT^{L22/L22}*; 3, *C155>DSiaT DSiaT^{L22/L22}*, rescue genotype. ^§, genotypes had matching genetic backgrounds that included *C155-Gal4* driver and were heterozygous for *para-GFP*. Ten to fifteen 7-day-old females were used per genotype in each experiment. For complete genotype information, see *Supplementary file 1*.

The online version of this article includes the following source data and figure supplement(s) for figure 9:

**Figure supplement 1.** Analyses of *para mRNA* and mCD8-GFP expression in *DSiaT* mutants, as controls for the specificity of the posttranscriptional effect of *DSiaT* on the level of Para-GFP.

**Figure supplement 1—source data 1.** Source data for *Figure 9—figure supplement 1A*.

**Figure supplement 1—source data 2.** Source data for *Figure 9—figure supplement 1B*.

**Figure supplement 1—source data 3.** Source data for *Figure 9—figure supplement 1C*.

**Source data 1.** Source data for *Figure 9A*.

**Source data 2.** Source data for *Figure 9B*.

studies suggested that NANS plays an essential role in the *Drosophila* sialylation pathway, which is consistent with the in vivo requirement of NANS for CSAS function revealed in our experiments.

A unique bipartite arrangement of the *Drosophila* sialylation pathway suggests that the pathway plays an important regulatory role in the nervous system. The separation CSAS and DSiaT functions between glial cells and neurons provides a mechanism for glia-mediated support of neuronal excitability via supplying neurons with glia-produced CMP-Sia, the sugar-nucleotide donor required as a substrate for DSiaT activity. Our data indicated that this mechanism could play regulatory functions. This scenario is consistent with the results indicating that CSAS mediates a rate-limiting step in the sialylation pathway, and thus changes in CSAS activity are expected to modulate neuronal excitability and affect neural transmission. Indeed, *CSAS* heterozygotes displayed a mild TS paralysis phenotype, revealing that a single functional copy of the *CSAS* gene is not sufficient for normal neural

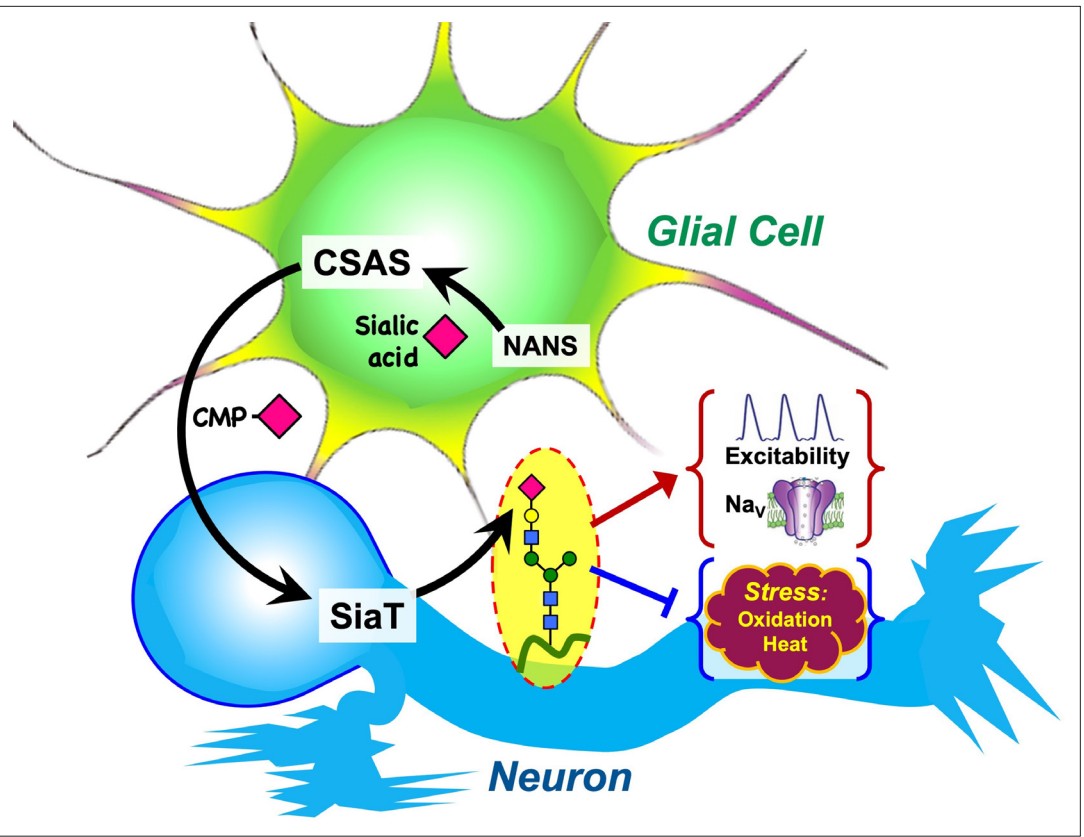

**Figure 10.** Graphical summary of the proposed mechanism of glia-neuron coupling via a bipartite sialylation pathway. In *Drosophila* brain, the last two steps of the sialylation pathway that are mediated by CMP-sialic acid synthetase (CSAS) and *Drosophila* sialyltransferase (DSiaT) are separated between glia and neurons. As a result, glia provides CMP-Sia to neurons that carry out sialylation of glycoproteins. This new mode of glia-neuron coupling promotes neural excitability, maintains the normal level of voltage-gated Na⁺ channels (Na$_V$), and counteracts the effects of heat and oxidative stress.

functions. At the same time, the overexpression of CSAS in wild-type flies increased their tolerance to heat above normal level (*Figure 7*). Taken together, our results suggested that the CSAS activity is a regulatory 'bottleneck' of the pathway. Notably, previous in vitro analyses revealed a uniquely steep dependence of CSAS enzymatic activity on temperature, demonstrating that the activity increases with temperature about an order of magnitude between 20°C and 40°C, a range of temperature of *Drosophila* natural habitats (*Mertsalov et al., 2016*). *Drosophila* is a poikilotherm, with its body temperature being regulated by ambient conditions, thus the changes in environmental temperature are predicted to modulate neural functions via the effect on CSAS activity. Coincidently with influencing heat tolerance, CSAS was also found to have a pronounced effect on the sensitivity to oxidative stress (*Figure 8*). CSAS overexpression in wild-type flies increased their survival in oxidative stress conditions, while CSAS deficiency made flies increasingly sensitive to oxidative stress, dramatically decreasing their survival. Although the molecular mechanism underlying the effect of sialylation on heat and oxidative stress tolerance remain to be elucidated, it is tempting to speculate that sialylation promotes the function of voltage-gated channels, such as the sodium channel Para, which leads to improved neuronal excitability and supports the stability of neural transmission under stress conditions. Mutations affecting Para commonly lead to temperature-induced paralysis (*Ganetzky, 1984*), and previous studies revealed strong synergistic interactions between *Para* and sialylation genes in producing this phenotype (*Repnikova et al., 2010*; *Islam et al., 2013*). Consistent with these results, we found that *DSiaT* function is required to maintain the normal level of Para expression (*Figure 9*). Generation of ROS is known to inhibit neuronal excitability (*Avshalumov et al., 2005*; *Pardillo-Díaz et al., 2015*; *Dantzler et al., 2019*), and thus the role of sialylation in promoting the function of Para is also consistent with the pronounced sensitivity of *CSAS* mutants to oxidative stress and the increased

oxidative stress tolerance of flies with *CSAS* upregulation (*Figure 6*). Vertebrate voltage-gated Na$^+$ channels are abundantly modified with sialylated glycans (*Miller et al., 1983*; *James and Agnew, 1987*; *Schmidt and Catterall, 1987*) that can affect channel gating and subcellular distribution in a context-dependent manner, while mutations causing defects in channel glycosylation are associated with heart and neurological disorders in humans and mice (*Montpetit et al., 2009*; *Jones et al., 2016*; *Cortada et al., 2019*), reviewed in *Ednie and Bennett, 2012*; *Scott and Panin, 2014*; *Fux et al., 2018*. The requirement of DSiaT for the normal level of voltage-gated Na$^+$ channels uncovered in our experiments suggests that similar mechanisms may operate in mammals. However, the glycans of invertebrate voltage-gated channels have not been analyzed and whether Para is a direct target of sialylation remains unknown. Characterizing glycosylation of Para and identifying molecular targets of sialylation in *Drosophila* will be important for elucidation of molecular mechanisms of the sialylation-mediated coupling between glia and neurons.

The mechanism of 'outsourcing' the production CMP-Sia to glia while downregulating the biosynthesis of Sia in neurons that consume CMP-Sia is consistent with metabolic differences between glia and neurons. Glial cells are more glycolytic and can provide metabolic support to neurons, e.g., by secreting lactate, while neurons, the cells with the highest demand for energy, generally rely on oxidative phosphorylation for energy production (although some studies indicated that neurons could also upregulate glycolysis during stimulation; *Diaz-García et al., 2017*; *Magistretti and Allaman, 2018*). Consistent with the scenario of different metabolic 'specialization', the neuron-specific silencing of glycolytic genes caused no abnormal phenotypes in flies, while the downregulation of glycolysis in glia resulted in neurodegeneration (*Volkenhoff et al., 2015*). Biosynthesis of Sia may compete with the energy metabolism of neurons by consuming pyruvate, which can potentially strain neurons as they use pyruvate as an essential source of energy. On the other hand, Sia production can be more readily supported in glial cells by their active glycolysis. It will be interesting to investigate if the sialylation-mediated glia-neuron coupling that we discovered in *Drosophila* is evolutionarily conserved in mammals. Unlike *Drosophila*, mammalian organisms apparently maintain all steps of the sialylation pathway ubiquitously active in all cells, with the brain being the organ with the highest concentration of Sia (*Schnaar et al., 2014*). It is conceivable that mammalian glial cells may promote neuronal sialylation by providing an exogenous supply of CMP-Sia, in addition to CMP-Sia produced by neurons, as the latter is possibly limited due to metabolic constrains and/or the inhibition of GNE (UDP-GlcNAc 2-epimerase/ManNAc kinase), a key regulatory mechanism that keeps mammalian sialylation in check (*Hinderlich et al., 2015*). Further studies are needed to test this intriguing possibility.

Our study raises an intriguing question about possible mechanisms of transferring CMP-Sia from glial cells to neurons. Although our experiments did not directly address this question, the function of CMP-Sia as a secreted diffusible factor is in agreement with our finding that the paralysis phenotype of *CSAS* mutants can be fully rescued by the transgenic expression of CSAS in different subtypes of glial cells, including ensheathing glia, astrocyte-like glia, and subperineurial glia (*Figure 2A*). Considering the localization of CSAS in the Golgi, it is tempting to speculate that CMP-Sia can be secreted via exocytosis, however other scenarios of CMP-Sia secretion and delivery to neurons are possible, including extracellular vesicles (*Inaba et al., 2022*). Further studies will focus on these mechanisms, which may shed light on analogous regulatory processes that operate in the mammalian nervous system.

## Materials and methods
### *Drosophila* strains

The following strains used in the study were obtained from the Bloomington Stock Center, Indiana University: *C155-Gal4* (#458), *repo-Gal4* (#7415), *1407-Gal4* (#8751), *Act-Gal4* (#3954), *C164-Gal4* (#33807), *UAS-DSiaT-RNAi* (#44528), *LexAop2-mCD8-GFP* (#32203), *LexAop-GFP.nls* (#29954), *LexAop-FLP* (#55820). *UAS-CSAS-RNAi* (#101396) was obtained from the Vienna Drosophila Resource Center. *Gli* and *MZ709 Gal4* lines were obtained from Michael Stern (Rice University), *Mj85b-Gal4* was from Josh Dubnau (Stony Brook University). *CSAS* and *DSiaT* mutant alleles and *UAS-DSiaT* and *UAS-CSAS* transgenic constructs were previously described (*Repnikova et al., 2010*; *Islam et al., 2013*). All mutant alleles were multiply outcrossed to $w^{1118}$ Canton S which was used as a 'wild-type' genetic background. Unless indicated otherwise, *Drosophila* strains were reared in a controlled environment

at 25°C in 60% humidity with 12 hr day/night light cycles. See *Supplementary file 1* for detailed information on genotypes used in experiments. Genetic strains created in this study are available per request.

## Transgenic constructs

*CSAS-LexA* driver line was generated using a BAC clone (CH322-158A02, CHORI P[acman] library [*Venken et al., 2009*], BACPAC Genomics, CA, USA) with ~22.1 kb genomic fragment including the *CSAS* ORF (0.97 kb) approximately in the middle. We used recombineering approach (*Venken et al., 2008*) to replace the *CSAS* coding region of the last exon with the sequence encoding LexA::p65 transcription activator (*Pfeiffer et al., 2010*). The resulting BAC construct included intact upstream, downstream, and intron sequences of the *CSAS ORF* region and thus predicted to express LexA::p65 in the endogenous *CSAS* pattern when introduced in vivo. BAC-DSiaT-HA was generated using a similar strategy. The BAC clone CH322-7B13 carrying ~20.1 kb genomic fragment with *DSiaT ORF* (2.97 kb) approximately in the middle was used in recombineering to introduce in-frame a 3xHA tag coding sequence at the 3' end of the *DSiaT* coding region. A short linker encoding 16 amino acids was included with the 3xHA tag sequence due to the presence of a *LoxP* site. The linker was previously shown not to significantly affect protein expression and localization (*Venken et al., 2008*). Outside of the tag insertion, *BAC-DSiaT-HA* contained unchanged genomic sequences and is expected to express a 3xHA-tagged DSiaT protein in the endogenous pattern in vivo. Transgenic *Drosophila* strains carrying *CSAS-LexA* and *DSiaT-HA* constructs were obtained by phiC31 integrase-mediated insertion (*Bischof et al., 2007*). We generated *UAS-NANS* construct by inserting *NANS* cDNA into the *pUASTattB* vector using standard molecular cloning techniques. A full-length *NANS* cDNA clone (IP20889) was obtained from the Drosophila Genomics Resource Center (Indiana University, Bloomington, IN, USA).

## Behavioral assays

Behavioral assays were performed essentially as described previously (*Repnikova et al., 2010*; *Islam et al., 2013*). Briefly, flies were collected on the day of eclosure and aged for 5 days, during which they were transferred once on day 3 to a fresh-food vial. For TS paralysis assays, unless indicated otherwise, individual flies were transferred to empty vials and the vials were submerged in a 38°C temperature-controlled water bath. We defined paralysis as a condition when a fly is down and unable to stand and walk for at least 1 min. At least 20 flies were assayed for each genotype. For righting assays, individual flies were placed in a vial and allowed to acclimate for 10 min. The vials were tapped on a soft foam pad five times twice, the time they spent on their back was recorded. Two trials for each fly with a 10 min time between trials were performed. Same-sex animals (females, unless indicated otherwise) were compared in any particular behavioral experiment. To decrease the effect of differences in genetic background, we used genotypes that were outcrossed to the same reference background ($w^{1118}$ Canton S) and/or siblings from the same parents. At least 20 flies were assayed for every genotype, unless indicated otherwise.

## Dissections and immunostaining

Brains were dissected in ice-cold Ringer's solution, washed, and fixed in fresh fixative solution (4% paraformaldehyde, 50 mM NaCl, 0.1 M Pipes, pH 7.2) for 20 min at room temperature with gentle agitation. Fixed tissues were analyzed by immunostaining and microscopy. Immunostaining was performed using fluorescent secondary antibodies essentially as described earlier (*Lyalin et al., 2006*). The following primary antibodies and corresponding dilutions were used for immunostaining: mouse anti-GFP 8H11 (1:100), anti-Repo 8D12 (1:10), rat anti-Elav 7E8A10 (1:10), mouse anti-Brp nc82 (1:10), all from Developmental Studies Hybridoma Bank; rabbit anti-GFP from Invitrogen (1:800), rat anti-HA from Roche (1:1000), rat anti-Dpn from Abcam (1:500). The following secondary antibodies were used: goat anti-rabbit and anti-mouse Alexa Fluor 546 and 488 (1:250), all from Invitrogen; donkey anti-mouse and anti-rabbit Cy3 (1:250) and FITC (1:150), from Jackson Laboratories. Stained samples were mounted on slides in Vectashield (Vector Laboratories) and imaged using Zeiss Axio Imager M2 fluorescence microscope with ApoTome module for optical sectioning or Zeiss 510 META Confocal microscope. Images were processed using Zeiss Zen and ImageJ software.

## Electrophysiology

Intracellular recordings were performed from NMJs of dissected third instar larvae essentially as previously described (*Islam et al., 2013*). Briefly, free-moving larvae were dissected in ice-cold $Ca^{2+}$-free HL3.1 buffer and EJP recordings were performed at 0.5 mM $Ca^{2+}$ at room temperature. EJPs were evoked by directly stimulating the segmental nerve innervating a hemisegment A3 Muscle 6/7 using a glass capillary electrode at 0.2 Hz with stimulus pulses of 0.3 ms duration. We recorded from a single muscle per animal, while collecting and averaging 20 EJPs from each larva. There were no differences in input resistance, time constant $\tau$, and resting membrane potential among different genotypes tested in the experiments shown, and the EJP amplitudes were corrected for nonlinear summation (*Martin, 1955*). Data were processed with Mini Analysis Program by Synaptosoft, Clampfit, and Excel.

## Oxidative stress experiments

To assess sensitivity to oxidative stress, we used a paraquat-induced stress paradigm as previously described (*Zou et al., 2000*) with some modifications. Briefly, flies were collected on the day of eclosure and aged in groups of 10 for 6–7 days on regular medium, while been transferred once to new vials on day 3. Before exposure to paraquat, flies were starved in empty vials for 6 hr, and then transferred into new vials containing 23 mm filter paper discs (Whatman G3) soaked with 40 mM paraquat in 5% sucrose, or 5% sucrose as a control. Flies were kept at a chamber with 100% humidity during starvation and paraquat exposure. They were transferred into new vials with fresh paraquat/sucrose every 48 hr. Mortality was assessed at defined time points as indicated. For statistical analyses, >10 independent experiments were performed for every genotype.

## Metabolomic analyses of CMP-Sia

Sample preparation and LC-MS analyses were carried out essentially as previously described with minor modifications (*Willems et al., 2019*). Briefly, lyophilized flies were homogenized using a glass potter with 400 µL of ice-cold 75 mM ammonium carbonate (Honeywell, Fluka) buffer adjusted to a pH of 7.4 with acetic acid. The homogenates were transferred to 1.5 mL Eppendorf tubes and snap-frozen in liquid nitrogen to be further stored in –80°C. Pierce BCA Protein Assay Kit (Thermo Fisher Scientific) was performed to each homogenate and three technical replicates were made using an equivalent of 270 µg protein content. Prior to metabolite extraction, 10 µM of $^{13}C3$-*N*-acetylneuraminic acid (Merck) was added as an internal standard to all replicates. Each sample was incubated with extraction solvent (2:2 acetonitrile/methanol [vol/vol]) for 5 min at –20°C. Samples were then centrifuged at 16,000 × *g* for 3 min followed by drying of the supernatant in a vacuum centrifuge at room temperature. The pellet was reconstituted in 100 µl of MilliQ.

Samples were analyzed using reverse-phase ion pair chromatography (Agilent Technologies 1290 Infinity) coupled to a triple-quadrupole mass spectrometer operating in negative ion mode (Agilent Technologies 6490 Triple Quad LC/MS). Chromatography was performed on an Acquity UPLC column (Waters, HSS T3 1.8 µm, 2.1×100 mm) using a gradient of mobile phase A (10 mM tributylamine, 12 mM acetic acid, 2 mM acetyl acetone, 3% MeOH in MilliQ) and mobile phase B (10 mM tributylamine, 12 mM acetic acid, 2 mM acetyl acetone, 3% MeOH in 50% acetonitrile and 50% MilliQ). The flow rate was 0.4 mL/min with a column temperature of 40°C and the injection volume was 2 µL per sample. CMP-Neu5Ac and several other metabolites were analyzed based on the indicated MRM transitions: CMP-Neu5Ac (*613.14->78.9 m/z*), UDP-HexNAc (*606.07->273 m/z*), Neu5Ac (*308.1->87 m/z*), glucosamine-6 phosphate (*258->79 m/z*), and KDN (3-deoxy-D-glycero-D-galacto-2-nonulosonic acid) (*267.08->87 m/z*). Peaks were annotated after comparison with peaks from commercial standards. For statistical analyses, data were obtained from three biological replicates, each including 100 adult flies (50 males plus 50 females) and analyzed using three technical repeats (two technical outliers were removed using the ROUT method; *Motulsky and Brown, 2006*). Data were plotted as a relative response to the internal standard ($^{13}C3$-*N*-acetylneuraminic acid). Data analysis was performed using Agilent MassHunter Quantitative Analysis software for peak integration and GraphPad Prism software.

## Glycomic analysis of N-linked glycans in *Drosophila* larval brains

N-glycan isolation and analyses were carried out essentially as previously described (*Aoki et al., 2007*; *Koles et al., 2007*). Briefly, third instar larvae were rinsed several times in ice-cold PBS and brains

were manually dissected on ice in PBS. After dissection, the brains were flash frozen in heptane on dry ice. Approximately 200 µl total volume of brains were collected for each genotype. Brains were homogenized and the resulting extracts were delipidated by organic solvent as previously described (*Aoki et al., 2007*). The resulting protein preparations were trypsinized and subjected to PNGaseF digestion to release N-glycans. Released N-glycans were analyzed as their permethylated derivatives by nanospray ionization mass spectrometry in positive ion mode. The permethylated N-glycans were dissolved in 50 µl of 1 mM sodium hydroxide in methanol/water (1:1) for infusion into an orbital ion trap mass spectrometer (Orbi-LTQ; Thermo Fisher Scientific) using a nanospray source at a syringe flow rate of 0.60 µl/min and capillary temperature set to 210°C. For fragmentation by collision-induced dissociation in MS/MS, a normalized collision energy of 35–40% was applied. Detection and relative quantification of the prevalence of individual N-glycans was accomplished using the total ion mapping (TIM) functionality of the Xcalibur software package version 2.0 (Thermo Fisher Scientific) as previously described (*Aoki et al., 2007*). For TIM, the m/z range from 600 to 2000 was automatically scanned in successive 2.8 mass unit windows with a window-to-window overlap of 0.8 mass units, which allowed the naturally occurring isotopes of each N-glycan species to be summed into a single response, thereby increasing detection sensitivity. Most N-glycan components were identified as singly, doubly, or triply charged, sodiated species (M+Na) in positive mode. Peaks for all charge states were summed for quantification. Graphic representations of N-glycan monosaccharide residues are consistent with the Symbol Nomenclature for Glycans (SNFG) as adopted by the glycobiology communities (*Neelamegham et al., 2019*). All raw mass spectrometric data were deposited at Glyco-Post (*Watanabe et al., 2021*), accession # GPST000260.

## Western blot analysis of Para-GFP

Samples for western blot analysis were prepared essentially as previously described (*Ravenscroft et al., 2020*). Briefly, 10–15 flies were homogenized in 1× Laemmli gel loading buffer (30 µL per fly), supplemented with 5 mM EDTA, 1 mM PMSF, and 1× protease inhibitor cocktail (Sigma). Insoluble material was removed by centrifugation 15 min at 18,000 × $g$, +4°C. Proteins were separated using 4–20% SDS-PAGE and transferred onto nitrocellulose membrane (Bio-Rad). Membrane was blocked in 5% non-fat dry milk in 1× TBST, pH 8.0 and developed using rabbit anti-GFP (1:4000, Thermo G10362) and goat anti-rabbit-HRP (1:9000, Jackson ImmunoResearch 111-035-0030) primary and secondary antibodies, respectively. For protein loading control, membranes were stained with Ponceau S prior immunostaining. SuperSignal Pico PLUS Chemiluminescent substrate (Thermo 34577) was used to develop the western blots. Chemiluminescent signal was recorded on GE/Amersham I600 imager and quantified using ImageJ.

## Quantitative RT-PCR analysis of *Para* expression

Total cellular RNA was isolated from 7-day-old adult flies using TRIzol reagent (Thermo Cat. #15596026) according to the manufacturer's protocol. Ten to fifteen flies were used in each experiment. Quantity and quality of isolated RNA was evaluated spectrophotometrically and with agarose gel-electrophoresis. cDNA was synthesized using 2 µg of total RNA by Maxima First Strand cDNA Synthesis Kit (Thermo Cat. #K1671). qRT-PCR was performed using PowerUp SYBR Green Master Mix (Thermo Cat. #A25741) on Bio-Rad CFX96 Real-Time PCR instrument. Ct values were determined with Bio-Rad's CFX Manager software. Relative expression of *para* was assessed by $2^{-\Delta\Delta Ct}$ method using α-tubulin as a control (*Livak and Schmittgen, 2001*; *Ponton et al., 2011*).

The following primers were used in these experiments:

PARAex10-F1 5'-ATGCGACGACGATTACGTGT-3'
PARAex10-R1 5'-GACAGGAAAGCCCATCCGAA-3'
α-Tubulin-F 5'-TGTCGCGTGTGAAACACTTC-3'
α-Tubulin-R 5'-AGCAGGCGTTTCCAATCTG-3'

## Experimental design and statistical analysis

All experiments were performed at least three times (biological replicates), unless indicated otherwise in text. Whenever it was possible, each experiment included at least three technical repeats. Unless indicated otherwise, data points shown in all figures represent different biological replicates.

Statistical analyses in experiments with multiple groups of data were performed by one-way ANOVA with Tukey post hoc comparisons. Survival curves were compared using log-range tests. In all figures, 1, 2, and 3 asterisks represent p values of <0.05, <0.01, and <0.001, respectively; NS indicates that no significant differences were found (p>0.05). Details on statistical analysis are included in figure legends, text, and supplementary materials. The sample size required for reliable statistical analyses was determined empirically, based on our previous experience and knowledge of the system and the assays. No power analysis was used to predetermine the sample size. GraphPad Prism software was used for statistical analyses.

## Acknowledgements

Stocks obtained from the Bloomington Drosophila Stock Center at Indiana University (NIH Grant P40-OD-018537) were used in this study. We thank Michael Stern and Josh Dubnau for providing *Drosophila* strains. We are grateful to Ajit Varki, Mark Zoran, Pamela Stanley, Paul Hardin, and Thomas Ravenscroft for valuable discussions of various parts of the project. We thank Dmitry Lyalin, Michiko Nakamura, and Natalie Solivan-Mejias for technical support.

## Additional information

### Funding

| Funder | Grant reference number | Author |
|---|---|---|
| National Institutes of Health | NS099409 | Vladislav Panin |
| National Institutes of Health | NS075534 | Vladislav Panin |
| TAMU-COANCYT | 2012-037(S) | Vladislav Panin |
| TAMU AgriLife IHA | | Vladislav Panin |
| National Institutes of Health | GM103490 | Michael Tiemeyer |
| Radboud Consortium for Glycoscience | | Dirk J Lefeber |

The funders had no role in study design, data collection and interpretation, or the decision to submit the work for publication.

### Author contributions

Hilary Scott, Data curation, Formal analysis, Investigation, Methodology, Validation, Visualization, Writing – original draft, Writing – review and editing; Boris Novikov, Data curation, Formal analysis, Investigation, Methodology, Validation, Visualization, Writing – review and editing; Berrak Ugur, Ilya Mertsalov, Project administration, Supervision, Formal analysis; Brooke Allen, Kazuhiro Aoki, Project administration, Supervision; Pedro Monagas-Valentin, Melissa Koff, Sarah Baas Robinson, Raisa Veizaj, Project administration; Dirk J Lefeber, Data curation, Funding acquisition, Investigation, Methodology, Project administration, Supervision, Writing – review and editing; Michael Tiemeyer, Data curation, Formal analysis, Funding acquisition, Investigation, Methodology, Project administration, Supervision, Writing – review and editing; Hugo Bellen, Writing – original draft, Data curation, Validation, Investigation, Supervision, Visualization, Formal analysis; Vladislav Panin, Conceptualization, Data curation, Formal analysis, Funding acquisition, Investigation, Methodology, Project administration, Supervision, Visualization, Writing – original draft, Writing – review and editing

### Author ORCIDs

Boris Novikov http://orcid.org/0009-0007-0458-8445
Berrak Ugur http://orcid.org/0000-0003-4806-8891
Hugo Bellen http://orcid.org/0000-0001-5992-5989
Vladislav Panin http://orcid.org/0000-0001-9126-1481

Decision letter and Author response
Decision letter https://doi.org/10.7554/eLife.78280.sa1
Author response https://doi.org/10.7554/eLife.78280.sa2

## Additional files

### Supplementary files
• Supplementary file 1. Supplementary table of genetic strains and transgenic constructs used in the study.

• MDAR checklist

### Data availability
All data generated or analysed during this study are included in the manuscript and supporting file; Source Data files have been uploaded to a public repository for Tables 1 and Supplementary Table 3.

The following dataset was generated:

| Author(s) | Year | Dataset title | Dataset URL | Database and Identifier |
|---|---|---|---|---|
| Tiemeyer M | 2023 | Larval brain in WT, CSAS, SiaT | https://glycopost.glycosmos.org/entry/GPST000260 | GlycoPOST, GPST000260 |

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
