## [Editor Report]

This important paper uses *Drosophila* as a model to study the sialylation pathway and its role in nervous system function. Intriguingly, the authors demonstrate that the final two steps of the sialylation biosynthetic pathway are split across glia (CSAS) and neurons (DSiaT). This compelling finding will interest a broad readership as it identifies a new mode by which glia support neuronal function.

---

## [Decision Letter]

**Decision letter after peer review:**

Thank you for submitting your article "Glia-neuron coupling via a bipartite sialylation pathway promotes neural transmission and stress tolerance in *Drosophila*" for consideration by *eLife*. Your article has been reviewed by 3 peer reviewers, including Vilaiwan Fernandes as the Reviewing Editor and Reviewer #1, and the evaluation has been overseen by Claude Desplan as the Senior Editor. The decision is to ask you to revise the manuscript. The following individual involved in the review of your submission has agreed to reveal their identity: Matthew S. Macauley (Reviewer #3).

Essential revisions:

1) Please clarify which cells in the larval brain express DSiaT by using markers neuronal and neuroblast markers to know if it is truly neuronal or also in progenitors as this result may impact whether there are developmental defects involved. You should also test where GNE is expressed in the larval and adult brain.

2) Please use other glial drivers to test the requirement for DSiaT. Only sub-perineurial and ensheathing glial drivers were tested (Figure 2). You should also try other glial-type-specific drivers such as for astrocyte-like, cortex, and perineurial?

3) To rule out developmental defects, use a temperature-sensitive Gal80 to restrict GAL4/UAS perturbations to adult stages only. Please also provide additional controls requested by Reviewers 1 and 2. See below for more detail.

4) Reviewers were concerned about the mass spec results. In particular that the data shown are borderline and since DSiaT and CSAS are expressed most strongly during the adult stage (according to FlyBase). Therefore please redo these experiments with adults (see detailed comments from reviewer 2).

5) Provide appropriate controls for the stress sensitivity experiments (Figure 6) as genetic background seems to be having a large effect. UAS-only controls are essential. See detailed comments from Reviewers 1 and 2.

6) Discuss how CMP-Neu5Ac might be transported from glia to neurons.

*Reviewer #1 (Recommendations for the authors):*

– It would be good to see where DSiaT expression is exactly by using markers such as Elav and Dpn in the larval brain. It will be important to know if it is truly neuronal or also in progenitors as the rescue and knockdown experiments were not restricted to adult stages only.

– Only sub-perineurial and ensheathing glial drivers were tested (Figure 2), please can the authors also try other glial-type-specific drivers such as for astrocyte-like, cortex, and perineurial?

– C155 Elav-Gal4 is known to also drive expression in neuroblasts, therefore it would be better to use *nsyb*-Gal4 instead or to use a Gal80ts to restrict expression to adults only. Ideally, a Gal80ts would also be used with Repo-Gal4.

– The experiments shown in Figure 6 seem to be very sensitive to the specific genetic background, UAS-only controls or ideally backcrossing to isogenize the genotype are required to be fully confident in these results.

*Reviewer #2 (Recommendations for the authors):*

It would be nice to test whether the expression of CSAS in cortex glial cells and astrocytes is also sufficient for phenotypic rescue.

Rescue experiments should be repeated using a Gal80ts element to separate whether sialylated proteins are required during development or whether they are needed in adult flies. The temporal expression profile shown in FlyBase would suggest the latter.

Likewise, the biosynthesis of sialylated N-glycans should be determined in adult (aged) flies.

The notion that the level of CSAS is directly linked to stress sensitivity needs to be toned down or better controls should be provided. In Figure 6 A it is shown that wild-type flies are paralyzed about 1500 seconds after heat shock, and flies expressing act-Gal4 are paralyzed 550 seconds after heat shock The conclusion would be that expression of Gal4 makes flies more suspectable to heat stress. This effect is reduced by adding the CSAS transgene to 700 seconds. Please show UAS-CSAS alone. It would be better to combine data in Fig6A and 6B into one graph (same scale!) and use the following controls: CSAS mut / +, act-Gal4; CSAS mut / CSAS mut, act-Gal4; wt, act-Gal4; wt / UAS-CSAS, act-Gal4.

As the authors indicated in their introduction, the neural cell adhesion protein NCAM is a particularly well-studied example of a poly-sialylated protein. This poses the question of whether the fly homolog Fas2 is a similarly sialylated protein in neurons but not in glial cells.

*Reviewer #3 (Recommendations for the authors):*

P18, L24: Could the authors please name the irrelevant protein used as a control, within this text. Was this protein selected because it is known to not be sialylated?

Figure 7: the red symbols are mixed up (circles in the legend should be a diamond). Also, suggest that panel B matches the colors in panel A.

I'm curious about the product of ManNAc-6-P, which is the substrate for NANS. The mammalian equivalent is GNE which produces ManNAc-6-P from UDP-GlcNAc. Is it known where this enzyme is expressed in neurons and glial cells?

---

## [Author Response]

Essential revisions:1) Please clarify which cells in the larval brain express DSiaT by using markers neuronal and neuroblast markers to know if it is truly neuronal or also in progenitors as this result may impact whether there are developmental defects involved. You should also test where GNE is expressed in the larval and adult brain.

We performed double immunofluorescent staining of DSiaT-HA with neuronal and neuroblast markers, Elav and Deadpan, respectively (new Supplementary Figure 3), which confirmed that DSiaT is expressed only in neurons but not in neuroblasts. This result is consistent with our previous analysis of DSiaT expression (Repnikova et al. 2010).

Homologues of mammalian GNE, UDP-GlcNAc 2-epimerase/ManNAc kinase, have not been identified in protostomes, including *Drosophila*. Several hypotheses that may explain this puzzling fact have been discussed, such as a potential separation of the two activities of the bifunctional GNE enzyme, UDPGlcNAc 2-epimerase and ManNAc kinase, between two distinct proteins with limited overall homology to GNE, as well as other possibilities (Koles et al., Glycoconj J 2009 26:313–324). These hypotheses, however, are speculative and how precursors of sialylic acid are synthesized in flies remains a fascinating but unanswered question. This is on our high-priority list of future research directions.

2) Please use other glial drivers to test the requirement for DSiaT. Only sub-perineurial and ensheathing glial drivers were tested (Figure 2). You should also try other glial-type-specific drivers such as for astrocyte-like, cortex, and perineurial?

We followed the advice of reviewers and tested additional Gal4 drivers, including astrocyte-specific, neuropile ensheathing and perineurial glia drivers, dEAAT1, R56F03, and R85G01, respectively. Our results show now that the expression of CSAS in different glial subtypes can fully rescue the phenotype, which is indistinguishable from the rescue by pan-glial expression, while the expression in perineurial glia results in a partial rescue (new Figure 3A). The most straightforward interpretation of these results appears to indicate that CMP-Sia can be produced by CSAS and delivered to neurons from any type of glial cells inside the brain, while the expression outside the main blood-brain barrier (maintained largely by subperineurial cells) is not very efficient, probably because of inefficient transport of CMP-Sia across the layer of subperineurial cells. This conclusion is consistent with the CSAS expression pattern that includes numerous glial cells within the brain, apparently encompassing all major types of glial cells. In fact, the partial rescue with the perineuriel driver is possibly explained by its expression in a few cortex and astrocyte glial cells inside the brain, which was reported in a recent paper (Weiss et al., 2022). Thus, taken together, our results adequately substantiate the main conclusion that CSAS is expressed and functionally active in many different glial cells in the CNS, but it is not endogenously expressed and cannot function in neurons.

3) To rule out developmental defects, use a temperature-sensitive Gal80 to restrict GAL4/UAS perturbations to adult stages only. Please also provide additional controls requested by Reviewers 1 and 2. See below for more detail.

We agree that it will be important to understand whether the phenotypes of sialylation mutants are caused by developmental defects or post developmental neurophysiological abnormalities, or a combination of these two pathogenic mechanisms. In our opinion, however, this investigation is a large project on its own that requires preparing a number of new genotypes and outcrossing them to the same genetic background. This project goes beyond the scope and the timeframe of the present study.

4) Reviewers were concerned about the mass spec results. In particular that the data shown are borderline and since DSiaT and CSAS are expressed most strongly during the adult stage (according to FlyBase). Therefore please redo these experiments with adults (see detailed comments from reviewer 2).

Detection of sialylation in *Drosophila* is known to be challenging even by sensitive mass spectrometry approaches because sialylated glycans are present in flies at an exceedingly low level (Aoki et al., 2007; Koles et al., 2007). We understand the rationale of reviewers suggesting to use adult flies for MS analysis based on FlyBase transcriptomic data. However, the expression level of glycogenes is not always a good predictor of the level of the glycan structures that they produce because of a multilayered regulation of glycosylation at the level of protein production, posttranslational modifications, cell organization/ enzyme subcellular localization, availability of substrates, and other factors (e.g., Young WWJ J Membr Biol. 2004 198, 1–13; Marathe et al. FASEB J. 2008 22(12): 4154–4167, etc.). Indeed, our analysis of DSiaT expression in the adult brain indicated that the expression is rapidly decreasing after eclosion due to posttranscriptional regulation (see Author response image 1). Based on our experience, the detection of sialylated glycans in adult flies is significantly more challenging than in larvae due to (1) relatively lower level of sialylated glycans (probably because DSiaT protein expression is actually decreased in adult flies as compared to larvae), (2) a higher complexity of the N-glycome at the adult stage that precludes detection without special enrichment steps (Koles et al. 2007 Glycobiology 17:1388-1403), and (3) significant levels of contaminating compounds in adult tissues, especially hexose polymeric species, that suppress detection of all but the most abundant N-glycans (Aoki et al. 2007 JBC 282: 9127-9142). Per reviewers’ request, however, we did try to analyze sialylated N-glycans in adult flies using our most sensitive MS approach, but unfortunately, we could not obtain reliable MS2 fragmentation to support glycan identification and quantification. To further support our analysis of larval glycans, we added more biological replicates which makes the conclusions of our mass spec experiments more reliable (new Figure 5C-D and Supplementary File 2). We hope that these additional data adequately address the reviewers’ concern about the mass spec analysis.

**Author response image 1. sa2fig1:** DSiaT expression in the adult brain is regulated via a posttranscriptional mechanism. (**A-B**) The expression of DSiaT in the brain is significantly decreased in 10-day-old flies (B) as compared to newly eclosed flies (A, 0-day-old). Arrows indicate the regions of elevated expression in the olfactory system (projection neurons) and optic lobes. Asterisks indicate the increased background signal in the older brain (B). Images were obtained using identical experimental conditions (i.e. dissection, immunostaining, and imaging). (**C**) Quantitative RTPCR analysis of *DSiaT mRNA* expression in adult brains using primers for the two known splicoforms (short and long). Absence of correlation with the level of DSiaT protein expression (A-B) indicates that the expression of DSiaT undergoes a posttranscriptional regulation. Unpublished data obtained by Kate Koles.

5) Provide appropriate controls for the stress sensitivity experiments (Figure 6) as genetic background seems to be having a large effect. UAS-only controls are essential. See detailed comments from Reviewers 1 and 2.

We added the requested UAS-only control (new Figure 7B).

6) Discuss how CMP-Neu5Ac might be transported from glia to neurons.

We added a brief discussion of possible mechanisms of CMP-Sia transport. We decided to not elaborate on this topic in greater detail to avoid speculative considerations.

Reviewer #1 (Recommendations for the authors):– It would be good to see where DSiaT expression is exactly by using markers such as Elav and Dpn in the larval brain. It will be important to know if it is truly neuronal or also in progenitors as the rescue and knockdown experiments were not restricted to adult stages only.

We performed the suggested experiments. They provided additional confirmation that DSiaT expression is truly neuronal (new Figure 2 —figure supplement 3), which is consistent with our previously published results (Repnikova et al., 2010)

– Only sub-perineurial and ensheathing glial drivers were tested (Figure 2), please can the authors also try other glial-type-specific drivers such as for astrocyte-like, cortex, and perineurial?

In addition to previously tested pan-glial (Repo-Gal4), ensheathing (Mz709-Gal4), and subperineurial (Gli-Gal4) drivers, we performed new rescue experiments with drivers specific for astrocytes (dEAAT1Gal4), perineurial (R85G01-Gal4) and neuropile ensheathing (R56F03) glia, which further confirmed our conclusion that CSAS is required and functional in glial cells but not in neurons. See also our response to editors’ comments above.

– C155 Elav-Gal4 is known to also drive expression in neuroblasts, therefore it would be better to use nsyb-Gal4 instead or to use a Gal80ts to restrict expression to adults only. Ideally, a Gal80ts would also be used with Repo-Gal4.

We have not detected DSiaT expression in neuroblasts using Dpn as a marker. DSiaT expression is only detected in Elav-positive cells, indicating that DSiaT is expressed only in neurons but not in neuroblasts (new Figure 2—figure supplement 3; see also Repnikova et al., 2010). Furthermore, the expression of CSAS is only detected in Repo-positive cells, while CSAS ectopic expression by C155-Gal4 cannot rescue the phenotype of CSAS mutants, even though, as mentioned by Reviewer, the expression is presumably also induced in this case in some neuroblasts. Taken together, these results indicate that the sialylation pathway is not active in neuroblasts.

We agree with Reviewer that the investigation of stage-specific requirements of the sialylation pathways, e.g., using the Gal80ts system, is an important research direction. However, this is a large project that requires preparing and outcrossing a number of new genotypes. This investigation is beyond the scope and the timeframe of the present study.

– The experiments shown in Figure 6 seem to be very sensitive to the specific genetic background, UAS-only controls or ideally backcrossing to isogenize the genotype are required to be fully confident in these results.

We added a UAS-only (aka “no-driver”) control to new Figure 7 (previously Figure 6), which strengthen our conclusions. We would like to emphasize that the genotypes used in these experiments were multiply (5-7x) outcrossed to the same wildtype background used in all our experiments, and we are fully confident that these results do not show a non-specific effect of the genetic background. Please see our detailed response to a similar comment of Reviewer 2 below.

Reviewer #2 (Recommendations for the authors):It would be nice to test whether the expression of CSAS in cortex glial cells and astrocytes is also sufficient for phenotypic rescue.

We performed additional rescue experiments using drivers specific for astrocytes (dEAAT1-Gal4) and perineurial (R85G01-Gal4) and neuropile ensheathing (R56F03) glia. Together with previous rescue experiments, they further strengthen our main conclusion that CSAS is required and functional in glial cells but not in neurons. See also our response to editors’ comments above.

Rescue experiments should be repeated using a Gal80ts element to separate whether sialylated proteins are required during development or whether they are needed in adult flies. The temporal expression profile shown in FlyBase would suggest the latter.

We agree that it will be important to understand the requirement of sialylation at different developmental stages. However, this new investigation is beyond the scope of the present study and it will diffuse the focus of the manuscript. Furthermore, this project will require a number of new genotypes to be prepared and outcrossed, which is a substantial task that goes beyond the timeframe of the present study.

Likewise, the biosynthesis of sialylated N-glycans should be determined in adult (aged) flies.

We attempted to characterize sialylated glycans in adult flies using our most sensitive mass spec approaches. However, the low level of sialylation, the increased complexity of the adult N-glycome, and significant levels of contaminating compounds (especially hexose polymeric species) found in adult tissues made this analysis exceedingly difficult and precluded a reliable quantification of sialylated glycans. Instead of the analysis of adult flies, we added more replicates of MS analysis of larval brains, which strengthened our conclusions. See also our responses to Reviewer 1 and Editors above.

The notion that the level of CSAS is directly linked to stress sensitivity needs to be toned down or better controls should be provided. In Figure 6 A it is shown that wild-type flies are paralyzed about 1500 seconds after heat shock, and flies expressing act-Gal4 are paralyzed 550 seconds after heat shock The conclusion would be that expression of Gal4 makes flies more suspectable to heat stress. This effect is reduced by adding the CSAS transgene to 700 seconds. Please show UAS-CSAS alone. It would be better to combine data in Fig6A and 6B into one graph (same scale!) and use the following controls: CSAS mut / +, act-Gal4; CSAS mut / CSAS mut, act-Gal4; wt, act-Gal4; wt / UAS-CSAS, act-Gal4.

Per Reviewer’s suggestion, we provided an additional UAS-CSAS alone control (aka “no-driver” control) in Figure 7B (previously Figure 6B), which strengthened our conclusions. Please note that the assays shown in this figure were performed at 40C, not at 38C that was used in Figure 7A and other experiments with sialylation mutants. In Figure 7B, we assayed the effect of ectopic upregulation of CSAS in wildtype flies. Wildtype flies are considered to be “not paralyzed” in our assays at 38C, as their time to paralysis approaches 20-30 min (~1500 sec). Based on our experience, the increase of heat tolerance above the “wildtype level” cannot be assayed at 38C because the time to paralysis becomes too long, beyond the resolution of the assay at this temperature. The exposure to heat shock for longer than 30 min starts to induce irreversible physiological changes that commonly result in death (flies never recover) instead of the temporal paralysis phenotype that we intend to analyze. For comparing paralysis of “wildtype genotypes” with or without CSAS overexpression, we used increased temperature (40C) to decrease the heat-shock exposure time (see also Nakamura et al. G3 2012. 2: 653-6). The assays shown in Figure 7A and 7B were performed at different temperatures, which is the reason of the difference in paralysis time between “wildtype” genotypes in these figures. The presence of a Gal4 driver alone does not make flies significantly more sensitive to heat. Combining the data of these two figures in one graph will be confusing for readers as these results were obtained at different experimental conditions (we emphasized this in the figure legend).

As the authors indicated in their introduction, the neural cell adhesion protein NCAM is a particularly well-studied example of a poly-sialylated protein. This poses the question of whether the fly homolog Fas2 is a similarly sialylated protein in neurons but not in glial cells.

We agree with the reviewer that identifying functional targets of sialylation in flies is a fascinating question. In fact, this is among our outmost priorities of current/ future studies. Fas2 is certainly an important candidate for being a carrier of sialylated glycans. In this respect, we would like to mention that Fas2 is not expected to be poly-sialylated. Polysialylation has not been reliably identified in protostomes. Furthermore, *Drosophila* apparently lacks the genetic capacity to synthesize polysialic acid. DSiaT is arguably the only sialyltransferase in flies, and it was shown to be a member of the family of animal ST6Gal enzymes that cannot synthesize polysialic acid (Koles at al. JBC 2004).

Reviewer #3 (Recommendations for the authors):P18, L24: Could the authors please name the irrelevant protein used as a control, within this text. Was this protein selected because it is known to not be sialylated?

The irrelevant protein control was mCD8-GFP transgenically induced in all neurons using C155 driver. This information is included in Figure 9 —figure supplement 1. We now also added this information to the main text on P18. This exogenous marker protein is not expected to be affected by glycosylation (see Supplemental Figure 8B-C), and it was used as a control for a non-specific effect on membrane proteins expressed in neurons of *DSiaT* mutants.

Figure 7: the red symbols are mixed up (circles in the legend should be a diamond). Also, suggest that panel B matches the colors in panel A.

We apologize for the mixed-up symbols. We fixed the problem and made the suggested changes to the colors in Figure 8 (previously, Figure 7).

I'm curious about the product of ManNAc-6-P, which is the substrate for NANS. The mammalian equivalent is GNE which produces ManNAc-6-P from UDP-GlcNAc. Is it known where this enzyme is expressed in neurons and glial cells?

Reviewer raised a very interesting question. The mechanism of ManNAc-6-P biosynthesis in protostomes remains unknown. Homologues of mammalian GNE have not been identified in protostomes, including *Drosophila*. Although several hypotheses have been discussed to explain how the biosynthesis of sialic acid can be initiated in flies, they remain speculative. See also our response to Editors’ comments above.